# FEDERATED BAYESIAN OPTIMIZATION BASED ON SECURE DISTRIBUTED GAUSSIAN PROCESSES

## ABSTRACT

Bayesian optimization (BO) is a powerful framework for tuning expensive black-box functions, yet existing federated BO (FBO) methods either expose private data or rely on noise-based privacy mechanisms that degrade performance. We introduce FBO-FedGP, the first secure FBO algorithm that combines multiparty homomorphic encryption with a distributed sparse Gaussian process surrogate. Each client computes and encrypts local summary statistics on a shared support set, which are then aggregated homomorphically by the server into a global surrogate without ever accessing raw data. Clients adaptively balance exploration of the global surrogate with exploitation of their local posteriors via a monotonically increasing mixing schedule, ensuring both efficiency and personalization. We provide theoretical analysis showing that FBO-FedGP achieves sublinear cumulative regret under standard kernel assumptions, with explicit bounds linking regret to the support-set approximation error. Extensive experiments on 20 synthetic benchmarks and several real-world tasks show that FBO-FedGP consistently outperforms state-of-the-art privacy-preserving FBO baselines, improvements are statistically significant under paired Wilcoxon tests with Holm correction. Our framework also includes a concrete HE instantiation with empirically verified 128-bit security and negligible numerical error, demonstrating that strong privacy can be achieved without sacrificing accuracy. This work establishes a practical and scalable solution for secure collaborative zeroth-order optimization in federated environments.[1]

## 1 INTRODUCTION

Bayesian optimization (BO) is a data-efficient strategy for optimizing expensive black-box functions and has found widespread use in real-world problems, including materials design and discovery, sensor networks, the financial industry, and experimental design (Frazier & Wang, 2016; Garnett et al., 2010; Gonzalvez et al., 2019; Lorenz et al., 2019). More recently, it has gained increasing popularity in machine learning, where it has been employed for reinforcement learning, hyperparameter tuning, and neural architecture search (Turchetta et al., 2020; Wu et al., 2020; Kandasamy et al., 2018). These applications highlight the versatility of BO as a general-purpose framework for optimizing expensive black-box objectives. However, applying BO in federated learning (FL) settings is challenging due to several reasons, including design of efficient surrogate models to reduce compute and communication costs, preserving local data privacy and accounting for the data heterogeneity in federated settings.

Federated Bayesian optimization (FBO) tackles zeroth-order optimization in FL with design of federated surrogate models to reduce computational complexity while keeping the local data secure (Wang et al., 2023). Each collaborating participant builds its own Gaussian process (GP) (Williams & Rasmussen, 1995) surrogate model and share either GP hyperparameters (Zhu et al., 2024) or transformed GP approximation parameters for federated optimization, thus reducing prohibitive computational overheads and protecting the raw local data. Differential privacy (DP) is widely used in secure FBO methods to add noise at each iteration to protect newly found solutions (Dai et al., 2021). Other approaches transform decision variables with a fixed matrix before outsourcing optimization to a trusted third-party server (Kharkovskii et al., 2020). A recent work uses compressed sensing to hide data without noise or transforms (Liu et al., 2024).

---

[1]Code will be released upon acceptance.

Despite these advances, the following key challenges remain. Firstly, selecting noise level for data perturbation to maximize performance while keeping the local data secure is a major challenge in DP-based methods. Secondly, the perturbed data points at each participant must keep the same relative distances as the original data, failing this, search for a global optimum becomes hard and degrade the FBO performance. The compressed sensing (Liu et al., 2024) method keeps relative distances among data points of a participant and across the federation, but uses an exact GP model which scales cubically with data and adds to the communication complexity of overall system. Thirdly, kernel based methods (Xu et al., 2021) such as radial basis function network models achieve privacy preservation by sharing the model parameters across federation, however they do not provide high-quality uncertainty estimates as provided by the GP models. Finally, the parametrized GP models such as random fourier features (RFF) (Dai et al., 2020) based approximation of GP models for FBO degrade the optimization performance by limiting the fitting ability of the GP models.

Homomorphic encryption (HE) (Gentry, 2009) enables computation directly on encrypted data, thereby preserving the privacy of each participant's local inputs without degrading model performance. In contrast to DP or compressed sensing, HE secures data through encryption rather than noise injection or transformation, ensuring that accuracy is maintained. However, applying HE to an exact GP surrogate is computationally prohibitive in practice. To mitigate this, a secure distributed GP (dGP) framework (Nawaz et al., 2024; Chen et al., 2012) is employed, which matches the predictive accuracy of an centralized GP approximation while reducing the computational burden to a level suitable for real-world federated deployments. The dGP relies on a compact support set to summarize each participant's data into local summaries defined by mean and covariance statistics. The location and size of the support set directly influence the approximation error of the dGP models. To efficiently construct a representative support set while maintaining security, a federated clustering protocol is adopted. A key challenge in federated settings is that data heterogeneity can slow down convergence during optimization. To address this, a mixing schedule is introduced, where global surrogate models are emphasized in the early stages to explore the search space broadly, while local surrogate models are prioritized in later stages to capture fine-grained differences across participants' data. This yields a FBO framework that integrates secure dGP surrogates, preserves predictive accuracy, ensures privacy with HE, and effectively accounts for data heterogeneity through a mixture of global and local surrogates.

### RELATION TO PRIOR WORK

A closely related line of work developed secure aggregation primitives for dGP summaries using multiparty HE and empirically validated the cryptographic pipeline and wall-clock performance in representative regimes (Nawaz et al. (2024)). Our paper builds on these primitives but departs in three key aspects, (i) we adapt the secure dGP surrogate specifically for federated BO by introducing an efficient single-shot federated clustering procedure to form compact inducing sets, (ii) we provide new approximation-to-regret analyses that translate support-set approximation error into explicit cumulative-regret bounds for mixed local/global acquisition schedules (Sec. 4), and (iii) we substantially broaden empirical evaluation by adding ablations (impact of support-set selection strategy, federation size, mixing schedules, convergence visualizations etc.), statistical tests (paired Wilcoxon, Holm-corrected) and concrete simulation-based security model and demonstrate practical performance federated settings.

In summary, the contributions of this paper are as follows,

• This paper presents the FBO algorithm termed *FBO-FedGP* that employs a secure dGP surrogate for privacy-preserving optimization of expensive black-box functions. We provide simulation-based threat model and privacy goals for the secure dGP framework and validate it through analytical and empirical analysis (Sec. 2, Appendix B).

• An informative support-set selection procedure based on single-shot federated clustering is proposed to summarize each participant's local data with privacy safeguards. This protocol reduces computation and communication overheads and improves the dGP's predictive accuracy by selecting compact, representative inducing points (Alg. 1, Sec. 4).

• The framework addresses data heterogeneity by combining global and local surrogate models through a probability-based mixing schedule. The schedule emphasizes global information during early iterations to encourage broad exploration and shifts toward local surrogates in later stages to capture client-specific structure, thereby mitigating bias from non-iid data (Alg. 2, Sec. 4).

• We prove sublinear cumulative regret bounds for GP-UCB when using the secure dGP surrogate and the proposed mixing schedule. The bounds explicitly characterize the approximation error $\hat{\Delta}$ as a function of support-set size $|S|$ (Thm. 3, App. C).

• We evaluate on 20 synthetic benchmarks and several real federated tasks, reporting statistically significant improvements over five baselines (Table 5), ablations on support-set size and number of clients (Appendix D), and a concrete HE instantiation showing negligible impact on surrogate accuracy. (See Table 1, Fig. 1–2 and Appendix D for full results.)

## 2 Secure Distributed Gaussian Processes

In (Nawaz et al., 2024) standard dGP model is extended to secure dGP by encrypting each participant's local summary under a global public key. As in dGP, first a common support set $\mathcal{S} \subset \mathcal{X}$ is selected. Each participant $i$ computes a local summary $\mathbf{L}_i \triangleq (\dot{\mathbf{m}}_\mathcal{S}^{(i)}, \dot{\mathbf{K}}_{\mathcal{S}\mathcal{S}}^{(i)})$, consisting of a mean vector $\dot{\mathbf{m}}_\mathcal{S}^{(i)}$ and a covariance matrix $\dot{\mathbf{K}}_{\mathcal{S}\mathcal{S}}^{(i)}$.

To protect $\mathbf{L}_i$, each participant encodes and encrypts its entries under a global public key $\overline{\mathbf{b}}$. Concretely, let $\mathbf{t}^{(i)}$ denote the plaintext encoding of $\mathbf{L}_i$, the ciphertext $\mathbf{T}^{(i)} = (\mathbf{t}_0^{(i)}, \mathbf{t}_1^{(i)})$ is computed as follows.

$$\mathbf{t}_0^{(i)} = \mathbf{v}^{(i)} \cdot \overline{\mathbf{b}} + \mathbf{t}^{(i)} + \mathbf{e}_0^{(i)} \pmod{q}, \quad \mathbf{t}_1^{(i)} = \mathbf{v}^{(i)} \cdot \mathbf{a} + \mathbf{e}_1^{(i)} \pmod{q}, \tag{1}$$

where $\mathbf{v}^{(i)}$ is a small mask, $\mathbf{e}_j^{(i)}$ are Gaussian noise vectors, $\mathbf{a}$ and $q$ are HE parameters (see Nawaz et al. (2024) for parameter details).

The server aggregates all encrypted summaries via a homomorphic aggregation to obtain the secure global summary $\mathbf{G}' \triangleq (\mathbf{G_0}, \mathbf{G_1}) = \sum_{i=1}^{M} \mathbf{T}^{(i)} \implies \left( \sum_i \mathbf{t}_0^{(i)}, \sum_i \mathbf{t}_1^{(i)} \right)$, and then the server collaborates with participants to decrypt $\mathbf{G}'$ into the plaintext global summary by a collaborative decryption process. The decrypted global summary is given by $\tilde{\mathbf{G}} \simeq \mathbf{G} \triangleq (\overline{\mathbf{m}}_\mathcal{S}, \overline{\mathbf{K}}_{\mathcal{S}\mathcal{S}})$, with mean $\overline{\mathbf{m}}_\mathcal{S} \triangleq \sum_{i=1}^{M} \dot{\mathbf{m}}_\mathcal{S}^{(i)}$ and covariance $\overline{\mathbf{K}}_{\mathcal{S}\mathcal{S}} \triangleq \mathbf{\Sigma}_{\mathcal{S}\mathcal{S}} + \sum_{i=1}^{M} \dot{\mathbf{K}}_{\mathcal{S}\mathcal{S}}^{(i)}$.

Since all local summaries remain encrypted under HE until the final joint decryption, no raw data or intermediate information on participants is exposed, preserving local privacy. The decrypted global summary $(\overline{\mathbf{m}}_\mathcal{S}, \overline{\mathbf{K}}_{\mathcal{S}\mathcal{S}})$ is algebraically identical to that of a centralized sparse GP on $\bigcup_i \mathcal{D}_i$, yet protects each participant's privacy by design.

Following the computation of the global summary, the predictive mean and covariance for unobserved points in $\mathcal{U}_i \subset \mathcal{X}$ is given as follows.

$$\widehat{\mathbf{m}}_{\mathcal{U}_i} \triangleq \mu_{\mathcal{U}_i} + \mathbf{\Sigma}_{\mathcal{U}_i \mathcal{S}} \overline{\mathbf{K}}_{\mathcal{S}\mathcal{S}}^{-1} \overline{\mathbf{m}}_\mathcal{S}, \quad \widehat{\mathbf{K}}_{\mathcal{U}_i \mathcal{U}_i} \triangleq \mathbf{\Sigma}_{\mathcal{U}_i \mathcal{U}_i} - \mathbf{\Sigma}_{\mathcal{U}_i \mathcal{S}} \left( \mathbf{\Sigma}_{\mathcal{S}\mathcal{S}}^{-1} - \overline{\mathbf{K}}_{\mathcal{S}\mathcal{S}}^{-1} \right) \mathbf{\Sigma}_{\mathcal{S}\mathcal{U}_i} \tag{2}$$

where $\mu_{\mathcal{U}_i}$, $\mathbf{\Sigma}_{\mathcal{U}_i \mathcal{U}_i}$, and $\mathbf{\Sigma}_{\mathcal{U}_i \mathcal{S}}$ represent mean function on unobserved local data, covariance among unobserved data points, and the cross-covariance among unobserved data and support-set respectively. For details see Appendix A.

## 3 Problem Formulation

We consider the task of maximizing an unknown function $f : \mathcal{X} \to \mathbb{R}$ over a compact domain $\mathcal{X} \subset \mathbb{R}^d$ in a federated setting. A set of $M$ clients collaboratively participates in the optimization process. Each client $i \in [M]$ possesses a private, local dataset $\mathcal{D}_i = \{(\mathbf{x}_i, y_i)\}$ of input-output pairs from the global function $f$, potentially observed with noise.

The goal is to find a global maximizer $\mathbf{x}^* \in \arg\max_{\mathbf{x} \in \mathcal{X}} f(\mathbf{x})$ through a sequential decision-making process over $T$ rounds. In each round $t$, a client is selected to evaluate a point $\mathbf{x}_t$, observes a noisy outcome $y_t = f(\mathbf{x}_t) + \epsilon_t$, and adds this data to its local dataset. The key challenge is to design a protocol that coordinates this process under the strict constraints of data privacy, where raw data $\mathcal{D}_i$ from any client $i$ must never leave the local device. Additionally, the protocol must also account for the statistical heterogeneity where local data distributions $p_i(\mathbf{x}, y)$ can be non-i.i.d. across clients, meaning the local datasets $\mathcal{D}_i$ may be drawn from different, client-specific distributions. Moreover, the proposed protocol must also be computationally as well as communication efficient.

---

**Algorithm 1** Inducing Points Selection: Local and Global Clustering

---

1: **procedure** LOCALCLUSTERING($i, \mathcal{D}_i, k^{(i)}, \varepsilon, T_{\max}$)
2:     **Client $i$:** Initialize $\{\boldsymbol{\theta}_r^{(i)}\}_{r=1}^{k^{(i)}}$ via K-means++
3:     **for** $t = 1$ to $T_{\max}$ **do**
4:         **for** $r = 1$ to $k^{(i)}$ **do**
5:             $\zeta_r^{(i)} \leftarrow \{j : \|\mathbf{x}_j^{(i)} - \boldsymbol{\theta}_r^{(i)}\|_2 \leq \|\mathbf{x}_j^{(i)} - \boldsymbol{\theta}_s^{(i)}\|_2, \forall s\}$
6:         **end for**
7:         **for** $r = 1$ to $k^{(i)}$ **do**
8:             $\boldsymbol{\theta}_r^{(i)} \leftarrow \mu(\zeta_r^{(i)})$
9:         **end for**
10:     **if** $\max_r \|\boldsymbol{\theta}_r^{(i),\text{new}} - \boldsymbol{\theta}_r^{(i),\text{old}}\|_2 < \varepsilon$ **then break**
11:     **end if**
12:     **end for**
13:     **return** $\vartheta^{(i)} = \{\boldsymbol{\theta}_1^{(i)}, \ldots, \boldsymbol{\theta}_{k^{(i)}}^{(i)}\}$
14: **end procedure**

15: **procedure** GLOBALCLUSTERING($\{\vartheta^{(1)}, \ldots, \vartheta^{(M)}\}, |\mathcal{K}|$)
16:     **Server:** $\mathcal{C} \leftarrow \bigcup_{i=1}^{M} \vartheta^{(i)}$                         $\triangleright$ Aggregate local centers
17:     Initialize $\mathcal{Q}$ with centers from random client
18:     **while** $|\mathcal{Q}| < |\mathcal{K}|$ **do**
19:         $\overline{\boldsymbol{\theta}} \leftarrow \arg\max_{\boldsymbol{\theta} \in \mathcal{C} \setminus \mathcal{Q}} \min_{\boldsymbol{q} \in \mathcal{Q}} \|\boldsymbol{\theta} - \boldsymbol{q}\|_2$
20:         $\mathcal{Q} \leftarrow \mathcal{Q} \cup \{\overline{\boldsymbol{\theta}}\}$                    $\triangleright$ Greedy farthest-first selection
21:     **end while**
22:     Run K-means on $\mathcal{C}$ with initialization $\mathcal{Q} \rightarrow \{\tau_1, \ldots, \tau_{|\mathcal{K}|}\}$
23:     **for** $r = 1$ to $|\mathcal{K}|$ **do**
24:         $\tilde{\boldsymbol{\theta}}_r \leftarrow \mu(\tau_r)$                        $\triangleright$ Final global inducing points
25:     **end for**
26:     **return** $\mathcal{Z} = \{\tilde{\boldsymbol{\theta}}_1, \ldots, \tilde{\boldsymbol{\theta}}_{|\mathcal{K}|}\}$
27: **end procedure**

---

The objective is to design a federated algorithm that, after $T$ evaluations, minimizes global cumulative regret, $R_T = \sum_{t=1}^{T}[f(\mathbf{x}^*) - f(\mathbf{x}_t)]$, while adhering to the constraints above. The performance of any feasible algorithm will be governed by its ability to efficiently aggregate information from heterogeneous clients into a global model under a finite communication budget and strong privacy guarantees.

## 4 THE FBO-FEDGP FRAMEWORK

We now present FBO-FedGP, a framework for federated BO that leverages a federated, sparse GP surrogate model secured by HE.

### 4.1 OVERVIEW AND INITIALIZATION

The FBO-FedGP protocol, outlined in Algorithm 2, operates over $M$ clients and a central server. The process begins with a one-time initialization phase which proceeds as follows. First, each client $i$ builds an initial private dataset $\mathcal{D}_i$ by evaluating a few random points from the domain $\mathcal{X}$ (Alg. 2, lines 2-3). Then, a common, compact set of inducing points $\mathcal{Z}$ ($|\mathcal{Z}| = |\mathcal{S}|$) is collaboratively chosen. Each client proposes points via a local clustering algorithm on its local data (Alg. 1), and the server aggregates these to form the global support set $\mathcal{Z}$ (line 6). Finally each client initializes its local sparse GP surrogate $\mathcal{GP}^{(i)}$ using its data $\mathcal{D}_i$ and the shared inducing points $\mathcal{Z}$.

This setup ensures all subsequent computations are based on a fixed, shared representation, enabling efficient and secure aggregation.

**Algorithm 2** FBO-FedGP

> **Input:** Domain $\mathcal{X}$, $M$ clients, support budget $|\mathcal{S}|$
> 1: **for** $i = 1, \ldots, M$ **in parallel do**           ▷ Initialization
> 2:   Sample $\{x_j^{(i)}\} \subset \mathcal{X}$; Evaluate $y_j^{(i)} = f(x_j^{(i)})$; Add to $\mathcal{D}_i$
> 3:   Run $\vartheta^{(i)} \leftarrow \text{LOCALCLUSTERING}(\mathcal{D}_i)$       ▷ Alg. 1 (Lines 1–14)
> 4:   Send $\vartheta^{(i)}$ to server.
> 5: **end for**
> 6: Server: $\mathcal{Z} \leftarrow \text{GLOBALCLUSTERING}(\{\vartheta^{(i)}\})$; $|\mathcal{Z}| \leq |\mathcal{S}|$    ▷ Alg. 1 (Lines 15–27)
> 7: Broadcast $\mathcal{Z}$ to all clients
> 8: **for** $i = 1, \ldots, M$ **do**
> 9:   Initialize local GP $\mathcal{GP}^{(i)}$ on $\mathcal{D}_i$, $\mathcal{Z}$
> 10:   Initialize mixing schedule $p^{(i)} \in (0, 1]^T$
> 11: **end for**
> 12: **for** $t = 1, \ldots, T$ **do**            ▷ Main Optimization Loop
> 13:   **for** $i = 1, \ldots, M$ **do**
> 14:    Sample $r \sim \text{Uniform}(0, 1)$
> 15:    **if** $r < p_t^{(i)}$ **then**           ▷ Local Step
> 16:     Get $(\mu^{(i)}(x), \sigma^{(i)}(x))$ from $\mathcal{GP}^{(i)}$
> 17:     $x_t^{(i)} \leftarrow \arg\max_x \text{AF}(x \mid \mu^{(i)}(x), \sigma^{(i)}(x))$
> 18:    **else**              ▷ Global Step
> 19:     Compute encrypted summary $\text{Enc}(\mathbf{L}_i)$
> 20:     Server: $\text{Enc}(\mathbf{G'}) \leftarrow \bigoplus_i \text{Enc}(\mathbf{L}_i)$    ▷ Homomorphic aggregation
> 21:     Clients & Server: Decrypt $\text{Enc}(\mathbf{G'}) \rightarrow \mathbf{G}$    ▷ Collaborative decryption
> 22:     Compute global posterior $(\hat{\mu}(x), \hat{\sigma}(x))$ from $\mathbf{G}$
> 23:     $x_t^{(i)} \leftarrow \arg\max_x \text{AF}(x \mid \hat{\mu}(x), \hat{\sigma}(x))$
> 24:    **end if**
> 25:    Evaluate $y_t^{(i)} = f(x_t^{(i)})$; $\mathcal{D}_i \leftarrow \mathcal{D}_i \cup \{(x_t^{(i)}, y_t^{(i)})\}$
> 26:    Update local GP $\mathcal{GP}^{(i)}$ with new data
> 27:   **end for**
> 28: **end for**

## 4.2 SECURE FEDERATED OPTIMIZATION LOOP

The core of the optimization occurs over $T$ rounds. The key innovation is a client-specific mixing schedule $p_t^{(i)} \in (0, 1]$ that determines the trade-off between using a local model (for fast, exploitation-heavy steps) and initiating a secure global aggregation (for exploration and consensus).

**Local Step (lines 14-17):** With probability $p_t^{(i)}$, client $i$ queries its *local* surrogate $\mathcal{GP}^{(i)}$ to select the next point $x_t^{(i)}$. This is efficient and private, requiring no communication. The acquisition function (AF)—e.g., UCB, EI, or a posterior sample for Thompson Sampling—is computed directly from the local posterior $(\mu^{(i)}(x), \sigma^{(i)}(x))$.

**Global Step (lines 18-23):** With probability $1 - p_t^{(i)}$, a *global* decision is made. This process involves *secure aggregation* where each client $i$ computes a local summary $\mathbf{L}_i$ and encrypts it using HE. The server homomorphically aggregates these into a global encrypted summary $\mathbf{G'}$ (line 20). Then the clients collaboratively decrypt $\mathbf{G'}$ to yield the plaintext global summary $\mathbf{G}$ in *collaborative decryption* step, subsequently each client uses $\mathbf{G}$ to compute the global sparse posterior $(\hat{\mu}(x), \hat{\sigma}(x))$ and then queries the acquisition function to select $x_t^{(i)}$.

Finally, each client evaluates its chosen point $x_t^{(i)}$ to obtain $y_t^{(i)} = f(x_t^{(i)}) + \epsilon$ and updates its local dataset $\mathcal{D}_i$ (line 25).

### 4.3 The Mixing Schedule: Balancing Privacy and Performance

The mixing schedules $p_t^{(i)}$ are a crucial mechanism for balancing the cost of secure global aggregation against the need for collaborative exploration. A well-designed schedule $p_t^{(i)}$ should be monotonically increasing, satisfying the following,

$$\sum_{t=1}^{\infty}(1 - p_t^{(i)}) = \infty \quad \text{and} \quad \sum_{t=1}^{\infty}(1 - p_t^{(i)})^2 < \infty. \tag{3}$$

The first condition ensures global aggregation happens infinitely often, preventing clients from converging to disparate local optima. The second ensures the variance of the aggregation process remains bounded.

A canonical choice that meets these criteria is $p_t^{(i)} = 1 - t^{-c}$ for $0 < c \leq 1$. In our experiments, we use $p_t = 1 - 1/\sqrt{t}$ ($c = 0.5$), which provides substantial global exploration in early rounds while asymptotically favoring communication-efficient local steps. This offers a practical tradeoff, i.e., frequent early collaboration to align models and identify promising regions, followed by focused local refinement. The schedule can be tuned per-client based on resource constraints and data heterogeneity.

### 4.4 Analytic Convergence Analysis of FBO-FedGP

Stating the mathematical notations and basic assumptions. Let $\mathcal{X} \subset \mathbb{R}^D$ be compact, $f : \mathcal{X} \to \mathbb{R}$, and observations $y_t = f(x_t) + \epsilon_t$ with $\epsilon_t$ $\sigma$-sub-Gaussian. Denote the cumulative regret $R_T = \sum_{t=1}^{T}\big(f(x^\star) - f(x_t)\big)$ and simple regret $r_T = \min_{1 \leq t \leq T}\big(f(x^\star) - f(x_t)\big)$, where $x^\star \in \arg\max_{x \in \mathcal{X}} f(x)$. Assume $f \in \mathcal{H}_k$ with $\|f\|_{\mathcal{H}_k} \leq B$, and let $\gamma_T$ denote the standard GP information gain up to $T$ queries. We adopt GP-UCB with exploration schedule $\beta_t$ that yields the usual high-probability confidence event.

**Assumption 1** (Regularity). Following are the basic regularity assumptions. (i) The $f \in \mathcal{H}_k$, $\|f\|_{\mathcal{H}_k} \leq B$, where noise is $\sigma$-sub-Gaussian. (ii) The acquisition is GP-UCB with parameter $\beta_t$. (iii) The sparse posterior $(\widehat{\mu}_t, \widehat{\sigma}_t)$ admits a uniform approximation error bound $\widehat{\Delta} = \sup_{x \in \mathcal{X}, t \geq 1}\big(|\widehat{\mu}_t(x) - \mu_t(x)| + |\widehat{\sigma}_t(x) - \sigma_t(x)|\big) < \infty$, which monotonically decreases with $|\mathcal{S}|$ under standard Nystrom bounds. (iv) Encrypted aggregation reproduces the plaintext global summary exactly (homomorphic exactness). (v) The information gain functions $\gamma_t^{\mathrm{g}}$ and $\gamma_t^{(i)}$ for the global and local models are concave functions of $t$. (This holds for common kernels like linear, SE, and Matérn.)

The regret due to error accumulated by the sparse GP approximations is given as follows.

**Lemma 2** (Sparse-UCB Regret). *Run GP-UCB using the* sparse *posterior* $(\widehat{\mu}_{t-1}, \widehat{\sigma}_{t-1})$ *with the same exploration schedule* $\beta_t$ *as in Lemma 6 (Appendix C.2). Under Assumption 1, on the event* $\mathcal{E}$ *of Lemma 6 the cumulative regret satisfies*

$$\widehat{R}_T = \sum_{t=1}^{T}\big(f(x^\star) - f(x_t)\big) = \mathcal{O}\Big(\sqrt{T\,\beta_T\,\gamma_T} + T\,\sqrt{\beta_T}\widehat{\Delta}\Big).$$

Finally, the cumulative regret of FBO-FedGP framework with sparse GP approximation and local/global mixing schedule is given as follows.

**Theorem 3** (FBO-FedGP Sublinear Cumulative Regret). *Under Assumption 1 and a mixing schedule* $(p_t)_{t \geq 1} \subset (0, 1)$, *the FBO-FedGP with GP-UCB acquisition satisfies*

$$R_T = \mathcal{O}\Big(\sqrt{T\,\beta_T\,\Gamma_T} + T\sqrt{\beta_T}\,\widehat{\Delta}\Big), \qquad \Gamma_T \triangleq \gamma_{\mathbb{E}[T_{\mathrm{g}}]}^{\mathrm{g}} + \sum_{i=1}^{M}\gamma_{\mathbb{E}[T_\ell^{(i)}]}^{(i)}. \tag{4}$$

*In particular,* $R_T/T \to 0$ *and the simple regret* $r_T \to 0$ *in probability as* $T \to \infty$ *provided* $\widehat{\Delta} = o(1/\sqrt{\beta_T})$.

Complete analyses and proofs are detailed in Appendix C.

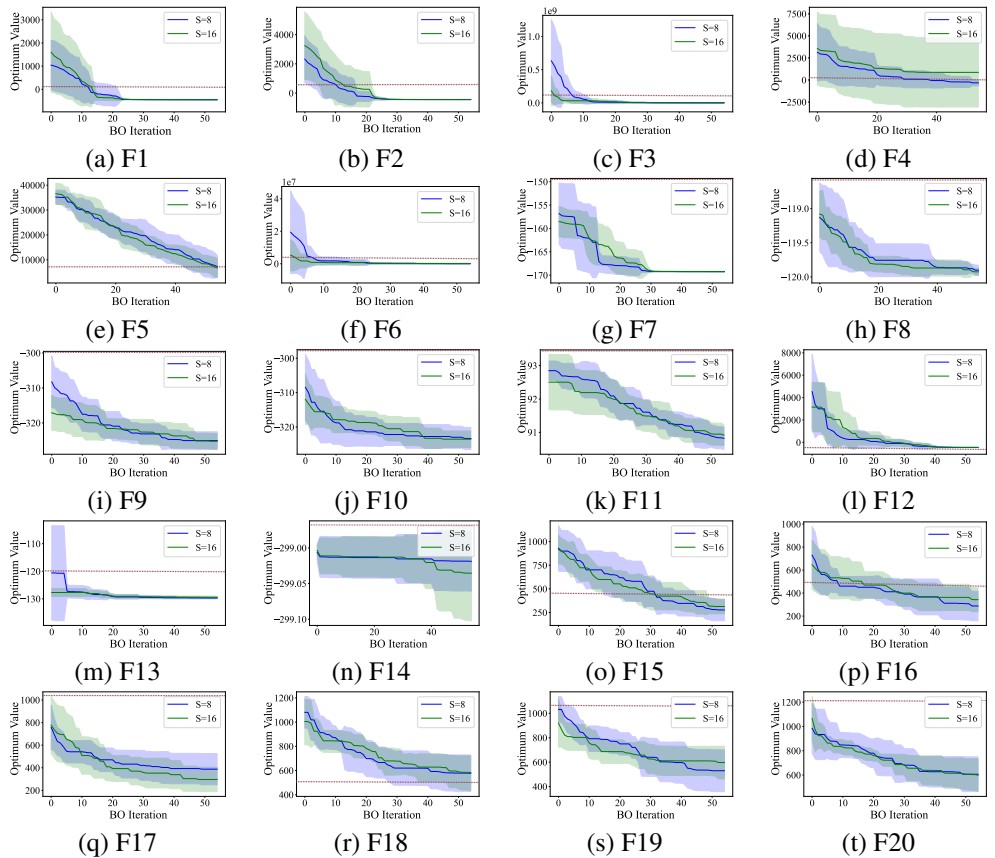

Figure 1: Solving F1–F20 using FBO-FedGP framework, the solid lines show the mean regret and the shaded regions show the std across the federation. The dotted red line shows the mean results obtained by competitive FBO frameworks listed in Table 1.

## 5 EXPERIMENTS

We evaluate FBO-FedGP on synthetic functions and real-world hyperparameter tuning tasks to assess its practicality and efficiency in a federated setting.

### 5.1 EXPERIMENTAL SETUP

The framework is implemented in Python. For synthetic functions, we simulate a federation of $M = 10$ clients. Each client starts with a small local dataset ($|\mathcal{D}_i| = 4$) and we run the optimization for $T = 50$ rounds using the GP-UCB acquisition function. All experiments are run on a uniform computing cluster for reproducibility. We use following two classes of benchmarks to evaluate different aspects of FBO-FedGP.

**Synthetic Benchmarks:** We use 20 single-objective functions from the CEC-2005 benchmark suite (Suganthan et al., 2005) with 10-dimensional input spaces. This diverse set includes unimodal and multimodal functions to test exploration and exploitation capabilities.

**Real-World Federated Learning Tasks:** We evaluate on realistic scenarios, including **UCI classification tasks** where we use three UCI datasets (Credit Default, Diabetes Health Indicators, and SUPPORT from UCI repository Asuncion & Newman (2007)) partitioned across clients using quantity-skewed partitioning ($\alpha = 10$) (Archetti et al., 2023) to simulate statistical heterogeneity and **cross-site landmine detection task** where we use the landmine detection dataset (Xue et al., 2007) with 29 distinct minefields (clients), emphasizing real-world heterogeneity and privacy requirements.

Table 1: Mean optimum values obtained over 10 independent runs compared with baseline methods. $+/\approx/-$ show wins, ties and losses of the algorithm compared to FBO-FedGP.

| Function | EPPBO | FMTBO | FTS | FEPPBO | FCSBO | FBO-FedGP |
|---|---|---|---|---|---|---|
| F1 | 27.33 − | 105.10 − | 143.14 − | 36.23 − | 3.32 ≈ | -268.49 |
| F2 | 122.97 − | 399.16 − | 726.49 − | 239.59 − | 21.81 + | 34.21 |
| F3 | 14200 − | 14300 − | 14100 − | 14100 − | 1.05E+8 − | 112.16 |
| F4 | 150.03 − | 476.92 − | 792.10 − | 236.45 − | 35.11 − | -183.78 |
| F5 | 7.02E+3 − | 1.29E+4 − | 1.69E+4 − | 1.26E+4 − | 1.57E+4 − | 1.12E+3 |
| F6 | 1.05E+5 − | 2.16E+5 − | 4.79E+5 − | 8.32E+5 − | 1.36E+5 − | 1.34E+3 |
| F7 | 76.99 − | 103.04 − | 103.63 − | -7.68 ≈ | 99.36 − | -169.28 |
| F8 | 10.50 − | 18.21 − | 32.14 − | 12.64 − | 9.93 − | -119.98 |
| F9 | -228.88 − | -228.01 − | -254.23 ≈ | -228.94 − | -252.28 ≈ | -328.56 |
| F10 | -208.79 − | -199.36 − | -221.58 − | -205.84 − | -209.26 − | -327.60 |
| F11 | 102.32 ≈ | 103.04 ≈ | 103.63 ≈ | 102.87 ≈ | 102.89 ≈ | 90.83 |
| F12 | 251.54 + | 723.56 − | 1145.03 − | 478.01 ≈ | 89.47 + | 459.30 |
| F13 | -69.41 − | -97.62 − | -97.62 − | -120.53 ≈ | -120.53 ≈ | -129.84 |
| F14 | -295.48 ≈ | -295.50 ≈ | -295.47 ≈ | -295.48 ≈ | -295.50 ≈ | -299.19 |
| F15 | 542.47 − | 876.89 − | 937.41 − | 632.21 − | 435.42 − | 120.42 |
| F16 | 573.59 − | 781.04 − | 1090.58 − | 714.91 − | 508.66 − | 169.79 |
| F17 | 1270 − | 1260 − | 1250 − | 1280 − | 1220 − | 734.39 |
| F18 | 520.89 − | 599.80 − | 587.70 − | 525.40 − | 563.47 − | 419.61 |
| F19 | 1270 − | 1260 − | 1250 − | 1280 − | 1220 − | 422.60 |
| F20 | 1450.98 − | 1560.32 − | 1670.54 − | 1434.49 − | 1250.01 − | 412.16 |
| $+/\approx/-$ | 1/2/18 | 0/2/18 | 0/3/17 | 0/5/15 | 2/7/13 | − |
| Ranking | 3.40 | 4.75 | 4.95 | 3.95 | 2.80 | 1.15 |

## 5.2 Baseline Methods and Evaluation Metrics

We compare against five privacy-preserving FBO methods representing different privacy approaches, including FEPPBO (Liu et al., 2024), EPPBO (Nguyen et al., 2018), FTS (Dai et al., 2021), FCSBO (Liu et al., 2024) and FMTBO (Zhu et al., 2024).

We assess performance using regret for synthetic functions, measuring convergence to optimum. We report classification loss for the classification tasks, measuring solution quality. Wall-clock times are reported to measure the computational and communication latency and performance over different acquisition strategies.

## 6 Results and Discussion

### 6.1 Performance on Synthetic Benchmarks

The proposed FBO-FedGP framework was evaluated on the CEC-2005 benchmark suite comprising 20 functions with diverse landscapes. Figure 1 shows convergence trajectories across all function categories.

On unimodal functions (F1–F5), FBO-FedGP demonstrated rapid convergence, with regret dropping sharply within the first 10 iterations. The framework showed particular robustness to ill-conditioning (F3) and noise (F4), confirming its strong exploitation capabilities in smooth landscapes. For multimodal functions (F6–F14), FBO-FedGP maintained steady improvement despite the presence of numerous local optima. While convergence was slower compared to unimodal functions, as expected for rugged landscapes, however the method consistently reduced regret over the optimization loop, effectively balancing exploration and exploitation. On the most challenging hybrid composition functions (F15–F20), FBO-FedGP achieved consistent convergence despite slower initial progress. The framework successfully adapted to non-separable landscapes and narrow basins, with the adaptive mixing schedule proving crucial for transitioning from global exploration to local exploitation.

Across all function types, FBO-FedGP achieved significant regret reduction, validating its ability to handle diverse optimization landscapes under federated constraints.

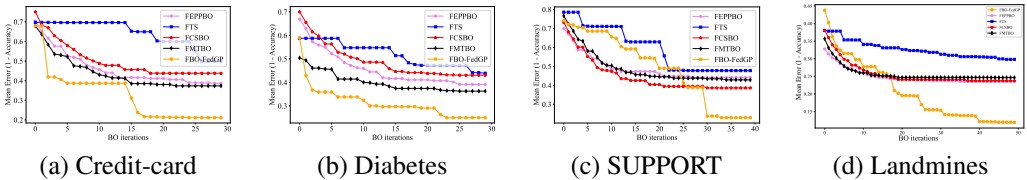

|        | (a) Credit-card | (b) Diabetes | (c) SUPPORT | (d) Landmines |
|--------|-----------------|--------------|-------------|---------------|

Figure 2: Convergence of FBO-FedGP framework on heterogeneous federated classification tasks.

## 6.2 COMPARISON WITH BASELINE METHODS

The FBO-FedGP was compared against five privacy-preserving FBO methods (EPPBO, FMTBO, FTS, FEPPBO, FCSBO) on synthetic benchmarks with 10 independent runs. As shown in Table 1, FBO-FedGP consistently achieved competitive or superior results. The method demonstrated particularly strong performance on complex multimodal functions, substantially outperforming baselines on F6 ($1.34 \times 10^3$ vs. baseline scores orders of magnitude higher) and F8 ($-119.98$). On F1, FBO-FedGP reached $-268.49$ compared to the best baseline at $3.32$. In cases where baselines showed comparable performance (F2, F12, F14), FBO-FedGP still delivered competitive results, indicating robust performance across diverse function characteristics. According to the Wilcoxon signed-rank test at a 0.05 significance level, FBO-FedGP achieves the best ranking across the synthetic benchmarks, consistently outperforming all baseline algorithms. FCSBO ranks second, showing closer competitiveness with FBO-FedGP, while EPPBO and FEPPBO obtain mid-level rankings. FMTBO and FTS are ranked lowest, indicating weaker optimization ability on these problems. These results demonstrate FBO-FedGP's capacity to effectively leverage global information while maintaining privacy, outperforming existing approaches across most benchmark problems.

## 6.3 REAL-WORLD CLASSIFICATION PERFORMANCE

The practical utility of FBO-FedGP was evaluated on four real-world classification tasks. Figure 2 shows convergence in terms of classification error (1 - accuracy).

On credit-card default prediction, FBO-FedGP achieved the fastest convergence, stabilizing near zero error within approximately 20 iterations, significantly outperforming all baselines. This demonstrates the framework's effectiveness in financial applications requiring both accuracy and rapid decision-making. Further, for diabetes classification, FBO-FedGP achieved the lowest final error with smooth convergence, while baselines showed fluctuations indicative of sensitivity to client-level data imbalance. This highlights the robustness of the FedGP surrogate under heterogeneous data distributions. Moreover, in the challenging in-hospital mortality prediction task, FBO-FedGP consistently outperformed alternatives, reaching near-zero error after 35 iterations while other methods plateaued at higher error levels. This demonstrates the method's ability to handle complex non-i.i.d. patterns in medical data under privacy constraints. Finally, on the landmine detection with significant cross-site heterogeneity, FBO-FedGP showed rapid error reduction within the first 15 iterations and achieved the lowest final error. The method avoided premature convergence that affected other approaches, demonstrating particular strength in heterogeneous, noisy environments. Across all real-world tasks, FBO-FedGP maintained superior performance while preserving privacy, confirming its practical utility in sensitive application domains.

For more experimental results refer to Appendix D.

## 7 CONCLUSION

We presented FBO-FedGP, a novel framework for federated BO that integrates sparse Gaussian processes with homomorphic encryption. Our method enables efficient and privacy-preserving optimization by leveraging a small set of support points for local summary computation and secure global aggregation. Theoretical analysis proves that FBO-FedGP achieves sublinear regret, while extensive experiments on synthetic and real-world benchmarks demonstrate that it outperforms existing FBO methods in terms of convergence speed and final solution quality, especially under heterogeneous data distributions. The framework provides a practical and theoretically grounded approach for privacy-sensitive optimization across distributed clients.

**Reproducibility statement.** To facilitate reproducibility we include an anonymized code and detailed appendices that together contain all material required to reproduce the experiments. The supplementary material contains an anonymized code snapshot which can be run using simple python command. For running the executables directly one may require to install and configure NTL (version 11.5.1), GMP (version 6.2.1) and HEAAN (version 2.1.0). Appendix A provides the definition of dGP models; Appendix B analyses the security of the dGP models and threat analysis; Appendix C gives the convergence analysis of FBO-FedGP framework and Appendix D gives supplementary experimental results and comparisons against the baselines.

**LLM Usage.** The ChatGPT was used to polish the presentation of several sections of the paper.

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

## A   DISTRIBUTED SPARSE GP (DGP) APPROXIMATION

Exact GP model over data $\mathcal{D}$ in input domain $\mathcal{X}$ is defined as $\mathcal{GP}(\mu(\cdot), c(\cdot, \cdot))$ where $\mu : \mathbb{R}^d \to \mathbb{R}$ is the mean function, and $c(\cdot, \cdot)$ is the positive-definite kernel function describing the covariances among the data points. The goal is to fit the GP on training data $\mathcal{D} \subset \mathcal{X}$ and perform inference on unobserved data $\mathcal{U} \subset \mathcal{X} \setminus \mathcal{D}$. To perform inference, the GP requires the following covariance matrices. $\Sigma_{\mathcal{DD}} \in \mathbb{R}^{n \times n}$, i.e., the covariance between training inputs, with entries $[\Sigma_{\mathcal{DD}}]_{i,j} = c(\mathbf{x}_i, \mathbf{x}_j)$ for $\mathbf{x}_i, \mathbf{x}_j \in \mathcal{D}$, $\Sigma_{\mathcal{UD}} \in \mathbb{R}^{m \times n}$, i.e., the cross-covariance between test and training inputs, with entries $[\Sigma_{\mathcal{UD}}]_{i,j} = c(\mathbf{x}'_i, \mathbf{x}_j)$ for $\mathbf{x}'_i \in \mathcal{U}$, $\mathbf{x}_j \in \mathcal{D}$, and $\Sigma_{\mathcal{UU}} \in \mathbb{R}^{m \times m}$, i.e., the covariance between test inputs, with entries $[\Sigma_{\mathcal{UU}}]_{i,j} = c(\mathbf{x}'_i, \mathbf{x}'_j)$ for $\mathbf{x}'_i, \mathbf{x}'_j \in \mathcal{U}$. The posterior mean and covariance of exact GP are given as follows.

$$\boldsymbol{\mu}_{\mathcal{U}}^{(\text{FGP})} = \boldsymbol{m}_{\mathcal{U}} + \Sigma_{\mathcal{UD}}(\Sigma_{\mathcal{DD}} + \sigma^2 \boldsymbol{I})^{-1}(\boldsymbol{y} - \boldsymbol{m}_{\mathcal{D}}),$$
$$\Sigma_{\mathcal{UU}}^{(\text{FGP})} = \Sigma_{\mathcal{UU}} + \sigma^2 \boldsymbol{I} - \Sigma_{\mathcal{UD}}(\Sigma_{\mathcal{DD}} + \sigma^2 \boldsymbol{I})^{-1}\Sigma_{\mathcal{DU}} \tag{5}$$

where $\boldsymbol{m}_{\mathcal{D}}$ and $\boldsymbol{m}_{\mathcal{U}}$ are mean functions on training and test data respectively, $\sigma^2$ is the variance of Gaussian noise $\epsilon \sim \mathcal{N}(0, \sigma^2)$.

It can be seen from equation 5 that exact GP computation requires the training data $\mathcal{D}$ to be centralized. However, in federated settings, the data is distributed among several participants and sharing of the local private data breaches the privacy requirements. Therefore, in federated setting, the dGP

models can be leveraged which compute a global GP model in a federated manner without sharing raw data. The distributed GP computation can be performed in two stages. Firstly, all $M$ participants independently compute a local summary using their private data $\mathcal{D}_i$ for $i = 1, \ldots, M$ and a commonly shared support set $\mathcal{S} \subset \mathcal{X}$ of size $|\mathcal{S}| \ll |\mathcal{D}|$. The local summary of a participant $\mathbf{L}_i \triangleq (\dot{\mathbf{m}}_{\mathcal{S}}^{(i)}, \dot{\mathbf{K}}_{\mathcal{SS}}^{(i)})$ comprises of a mean vector and a covariance matrix defined as follows.

$$\dot{\mathbf{m}}_{\mathcal{S}}^{(i)} \triangleq \boldsymbol{\Sigma}_{\mathcal{SD}_i} \boldsymbol{\Sigma}_{\mathcal{D}_i \mathcal{D}_i | \mathcal{S}}^{-1} \left( \boldsymbol{y}_{\mathcal{D}_i} - \boldsymbol{\mu}_{\mathcal{D}_i} \right), \tag{6}$$

$$\dot{\mathbf{K}}_{\mathcal{SS}}^{(i)} \triangleq \boldsymbol{\Sigma}_{\mathcal{SD}_i} \boldsymbol{\Sigma}_{\mathcal{D}_i \mathcal{D}_i | \mathcal{S}}^{-1} \boldsymbol{\Sigma}_{\mathcal{D}_i \mathcal{S}} \tag{7}$$

where $\mu_{\mathcal{D}_i}$ comprises of the expected values of outputs in $\mathcal{D}_i$, $\boldsymbol{\Sigma}_{\mathcal{D}_i \mathcal{D}_i | \mathcal{S}}^{-1}$, $\boldsymbol{\Sigma}_{\mathcal{SD}_i}$ and $\boldsymbol{\Sigma}_{\mathcal{D}_i \mathcal{S}}$ denote covariance and cross-covariance matrices.

In the second stage, a server aggregates these local summaries into a global summary $\mathbf{G}$ such that $\mathbf{G} \triangleq (\overline{\mathbf{m}}_{\mathcal{S}}, \overline{\mathbf{K}}_{\mathcal{SS}})$ for

$$\overline{\mathbf{m}}_{\mathcal{S}} \triangleq \sum_{i=1}^{M} \dot{\mathbf{m}}_{\mathcal{S}}^{(i)}, \quad \overline{\mathbf{K}}_{\mathcal{SS}} \triangleq \boldsymbol{\Sigma}_{\mathcal{SS}} + \sum_{i=1}^{M} \dot{\mathbf{K}}_{\mathcal{SS}}^{(i)} \tag{8}$$

where $\dot{\mathbf{m}}_{\mathcal{S}}^{(i)}$ and $\dot{\mathbf{K}}_{\mathcal{SS}}^{(i)}$ are the elements of local summary $\mathbf{L}_i$ as defined in equation 6 and equation 7 and $M$ is the number of collaborating participants.

Given the global summary $\mathbf{G}$ and the support set $\mathcal{S}$, the predictive Gaussian distribution for distributed partially independent training conditional (dPITC) approximation is $\mathcal{N}(\widehat{\mathbf{m}}_{\mathcal{U}_i}, \widehat{\mathbf{K}}_{\mathcal{U}_i \mathcal{U}_i})$ for unobserved points in $\mathcal{U}_i \subset \mathcal{X}$, where

$$\widehat{\mathbf{m}}_{\mathcal{U}_i} \triangleq \mu_{\mathcal{U}_i} + \boldsymbol{\Sigma}_{\mathcal{U}_i \mathcal{S}} \overline{\mathbf{K}}_{\mathcal{SS}}^{-1} \overline{\mathbf{m}}_{\mathcal{S}}, \tag{9}$$

$$\widehat{\mathbf{K}}_{\mathcal{U}_i \mathcal{U}_i} \triangleq \boldsymbol{\Sigma}_{\mathcal{U}_i \mathcal{U}_i} - \boldsymbol{\Sigma}_{\mathcal{U}_i \mathcal{S}} \left( \boldsymbol{\Sigma}_{\mathcal{SS}}^{-1} - \overline{\mathbf{K}}_{\mathcal{SS}}^{-1} \right) \boldsymbol{\Sigma}_{\mathcal{SU}_i} \tag{10}$$

where $\mu_{\mathcal{U}_i}$, $\boldsymbol{\Sigma}_{\mathcal{U}_i \mathcal{U}_i}$, $\boldsymbol{\Sigma}_{\mathcal{SS}}^{-1}$ and $\boldsymbol{\Sigma}_{\mathcal{U}_i \mathcal{S}}$ denote the mean vector and cross-covariance matrices respectively and $\mathbf{K}_{\mathcal{SU}_i}$ is the transpose of $\mathbf{K}_{\mathcal{U}_i \mathcal{S}}$, whereas $\overline{\mathbf{K}}_{\mathcal{SS}}^{-1}$ and $\overline{\mathbf{m}}_{\mathcal{S}}$ are defined in equation 8. For detailed definitions refer to (Nawaz et al., 2024; Chen et al., 2012).

# B    SECURITY ANALYSIS OF SECURE dGP

The secure dGP framework is resistant against the collusion attack by collaborating participants. For security analysis and the security parameters please refer to (Nawaz et al., 2024). Below we summarize the adversary model and privacy goals.

### ADVERSARY MODEL

A semi-honest (honest-but-curious) adversary model is adopted with the following characteristics.

- Any subset of participants (including the aggregator/server) may adhere to the prescribed protocol while potentially attempting to infer additional information from the exchanged messages.
- The adversary may collude with up to $c$ other participants, and explicitly consider the strong setting where $c = M - 1$.
- The adversary has knowledge of the public scheme parameters, i.e., the kernel $k(\cdot, \cdot)$, the support set $\mathcal{S}$, and any encoding/quantization parameters, and may possess auxiliary public data about the input domain.

### PRIVACY GOALS

This section details the privacy properties that th framework is designed to achieve in environments with semi-honest and colluding adversaries.

**Definition 4** (Simulation-Based Security). A protocol $\Pi$ is said to be secure for $M$ clients and a server $\mathsf{S}$ computing a dGP model if the following conditions are satisfied.

- **Correctness**. For all devices and the server participating in the computation of the dGP model, correctness ensures that for any set of client inputs $x^{(1)}, \ldots, x^{(M)}$ and server input $y$, the protocol outputs the correct global model $f(x^{(1)}, \ldots, x^{(M)}, y)$ with probability 1. Moreover, decryption of the final model parameters succeeds only if all clients contribute to the decryption process, ensuring robustness against missing or non-participating clients.

- **Security against a Semi-Honest Server**. For a semi-honest server $\mathsf{S}$, there exists a polynomial-time ideal simulator $\mathsf{Sim}_{\mathsf{S}}$ such that for any client inputs $x^{(1)}, \ldots, x^{(M)}$, public parameters $\lambda$, and server input $y$:

$$\mathsf{view}_{\mathsf{S}}(x^{(1)}, \ldots, x^{(M)}, \lambda, y) \overset{c}{\approx} \mathsf{Sim}_{\mathsf{S}}(y), \tag{11}$$

  where $\mathsf{view}_{\mathsf{S}}$ denotes the server's view during protocol execution. Eq. equation 11 guarantees that the server can only observe encrypted data and its own inputs, without learning any client's private data.

- **Security against Malicious Clients**. For any malicious client $\mathsf{C}^{(m)}$ deviating arbitrarily from $\Pi$, there exists a polynomial-time ideal simulator $\mathsf{Sim}_{\mathsf{C}^{(m)}}$ such that for the client's local input $x^{(m)}$:

$$\mathsf{view}_{\mathsf{C}^{(m)}}(x^{(m)}) \overset{c}{\approx} \mathsf{Sim}_{\mathsf{C}^{(m)}}(f(x^{(m)}, y)), \tag{12}$$

  where $f(\cdot, \cdot)$ denotes the global model output restricted to that client's input and public parameters. Eq. equation 12 ensures that a malicious client learns nothing beyond its own data and the intended global output.

- **Security against Collusion**. The protocol $\Pi$ is resistant to collusion between the server and up to $M - 1$ clients, meaning that the combined view of colluding parties reveals no information about honest clients' data beyond what can be inferred from the global output. Formally, for any subset of colluding adversaries $\mathcal{A} \subseteq \{\mathsf{S}, \mathsf{C}^{(1)}, \ldots, \mathsf{C}^{(M)}\}$ with $|\mathcal{A}| \leq M - 1$, there exists a polynomial-time ideal simulator $\mathsf{Sim}_{\mathcal{A}}$ such that:

$$\mathsf{view}_{\mathcal{A}}(x^{(1)}, \ldots, x^{(|\mathcal{A}|)}, y) \overset{c}{\approx} \mathsf{Sim}_{\mathcal{A}}(f(x^{(1)}, \ldots, x^{(|\mathcal{A}|)}, y)), \tag{13}$$

  where $\mathsf{view}_{\mathcal{A}}$ is the joint view of all colluding parties. Eq. equation 13 ensures that even a majority collusion (all but one client) cannot recover additional information about honest parties' inputs.

THEORETICAL ANALYSIS

In the following, analysis of dGP framework's security against collusion attacks involving up to $M - 1$ participants is provided.

The collaborative decryption mechanism in dGP proceeds as follows. Let the encrypted global summary be denoted by $\mathbf{G}' = (\mathbf{G}_0, \mathbf{G}_1)$. The plaintext global summary $\tilde{\mathbf{G}}$ is obtained by

$$\tilde{\mathbf{G}} \triangleq \mathbf{G}_0 + \Gamma \pmod{q}, \quad \Gamma = \sum_{i=1}^{M} \Gamma^{(i)}, \quad \Gamma^{(i)} = \mathbf{s}^{(i)} \cdot \mathbf{G}_1 + \bar{\mathbf{e}}, \tag{14}$$

where $\mathbf{s}^{(i)}$ is the private (secret) key of participant $i$, and $\bar{\mathbf{e}}$ denotes small error terms.

Each participant $i$ generates ciphertext components:

$$\mathbf{t}_0^{(i)} \triangleq \mathbf{v}^{(i)} \cdot \overline{\mathbf{b}} + \mathbf{t}^{(i)} + \mathbf{e}_0^{(i)} \pmod{q}, \tag{15}$$

$$\mathbf{t}_1^{(i)} \triangleq \mathbf{v}^{(i)} \cdot \mathbf{a} + \mathbf{e}_1^{(i)} \pmod{q}, \tag{16}$$

where $\mathbf{v}^{(i)}$ is a randomness vector, $\mathbf{t}^{(i)}$ is the encoded local summary, and $\mathbf{e}_0^{(i)}, \mathbf{e}_1^{(i)}$ are small noise terms sampled from discrete Gaussian distribution.

The global ciphertext components are shown as

$$\mathbf{G}_0 = \sum_{i=1}^{M} \mathbf{t}_0^{(i)}, \tag{17}$$

$$\mathbf{G}_1 = \sum_{i=1}^{M} \mathbf{t}_1^{(i)}. \tag{18}$$

Expanding Eq. equation 17 and Eq. equation 18 using Eqs. equation 15–equation 16,

$$\mathbf{G}_0 = -\sum_{i=1}^{M} v^{(i)} s^{(i)} \mathbf{a} + \sum_{i=1}^{M} v^{(i)} \mathbf{e}_0 + \sum_{i=1}^{M} \mathbf{t}^{(i)} + \sum_{i=1}^{M} \mathbf{e}_0^{(i)}, \tag{19}$$

$$\mathbf{G}_1 = \sum_{i=1}^{M} v^{(i)} s^{(i)} \mathbf{a} + \sum_{i=1}^{M} v^{(i)} \mathbf{e}_1 + \sum_{i=1}^{M} \bar{\mathbf{e}}. \tag{20}$$

Substituting Eq. equation 19 and Eq equation 20 into Eq. equation 14 gives,

$$\tilde{\mathbf{G}} = \left( -\sum_{i=1}^{M} v^{(i)} s^{(i)} \mathbf{a} + \sum_{i=1}^{M} \mathbf{t}^{(i)} + \text{noise} \right) + \sum_{i=1}^{M} s^{(i)} \left( \sum_{j=1}^{M} v^{(j)} s^{(j)} \mathbf{a} + \text{noise} \right) \quad (\text{mod } q).$$

The terms $\pm v^{(i)} s^{(i)} \mathbf{a}$ cancel exactly *only* when all $M$ participants contribute their decryption shares. If any share (say, from participant $k$) is missing, the residual term $-v^{(k)} s^{(k)} \mathbf{a}$ remains, leaving a large, secret-dependent mask that prevents recovery of $\sum_{i=1}^{M} \mathbf{t}^{(i)}$. This property holds under the RLWE assumption, ensuring that without all $M$ shares, the decryption output is computationally indistinguishable from random.

This implies that even with collusion of all but one participants, the missing share's secret key component guarantees that the plaintext global summary cannot be reconstructed thus establishing resilience against collusion attack and by corollary, the inference attack against any member of federation.

EXPERIMENTAL VALIDATION (COLLUSION)

We empirically validate that missing decryption shares prevent recovery of the global summary. When one share is intentionally withheld the attempted plaintext reconstruction exhibits modular wrap-around and produces values with extremely large magnitudes (due to interpreting a random modular residue in the plaintext domain). For example, a sample decrypted vector without the missing share contains entries with magnitudes on the order $10^{230}$,

$$\texttt{Decrypted Result:} \quad 1.73 \times 10^{230}, \; 9.26 \times 10^{230}, \; \dots$$

These values are astronomically large compared to original plaintext values, validating that the protocol outputs computationally meaningless data in the absence of all shares. This empirical observation aligns with the formal argument, confirming the framework's robustness.

**HE Parameters** We set the polynomial modulus degree and ciphertext modulus size as,

$$N = 2^{17} = 131{,}072, \qquad q = 2^{800} \approx 6.668 \times 10^{240},$$

and use a discrete Gaussian noise parameter $\sigma = 3.2$ and secret-key Hamming weight $h = 64$. The implementation also uses the internal primes and chain sizing parameters, which determine the exact modulus-chain decomposition used for CKKS-style arithmetic. All modulus/noise parameters are chosen such that the accumulated noise across the homomorphic operations in our aggregation pipeline remains well below the decryption bound, this is supported by our empirical error measurements (see below) and by the modulus-chain sizing in the implementation.

**Exactness of HE Operations** We measured the numeric error between secure and plaintext aggregation under full decryption: $\|\tilde{\mathbf{G}} - \mathbf{G}\|_\infty$ is $5.29 \times 10^{-7}$ for the global mean and $1.45 \times 10^{-11}$ for the global covariance, demonstrating negligible accuracy loss due to encoding/encryption in our parameter regime.

## C  Convergence Proofs for FBO-FedGP Framework

This appendix provides step-by-step proofs of the information-theoretic regret guarantees for the proposed FBO-FedGP with GP-UCB acquisition and SE kernel, following the framework of (Srinivas et al., 2012). Here we analyze the (standard) maximization setting, however, the minimization case follows by replacing $f$ with $-f$.

Stating the mathematical notations and basic assumptions. Let $\mathcal{X} \subset \mathbb{R}^D$ be compact, $f : \mathcal{X} \to \mathbb{R}$, and observations $y_t = f(x_t) + \epsilon_t$ with $\epsilon_t$ $\sigma$-sub-Gaussian. Denote the cumulative regret $R_T = \sum_{t=1}^{T} \big( f(x^\star) - f(x_t) \big)$ and simple regret $r_T = \min_{1 \le t \le T} \big( f(x^\star) - f(x_t) \big)$, where $x^\star \in \arg\max_{x \in \mathcal{X}} f(x)$. Assume $f \in \mathcal{H}_k$ with $\|f\|_{\mathcal{H}_k} \le B$, and let $\gamma_T$ denote the standard GP information gain up to $T$ queries. We adopt GP-UCB with exploration schedule $\beta_t$ that yields the usual high-probability confidence event.

### C.1  Assumptions (Restated)

**Assumption 5** (Regularity). Following are the basic regularity assumptions. (i) The $f \in \mathcal{H}_k$, $\|f\|_{\mathcal{H}_k} \le B$, where noise is $\sigma$-sub-Gaussian. (ii) The acquisition is GP-UCB with parameter $\beta_t$. (iii) The sparse posterior $(\widehat{\mu}_t, \widehat{\sigma}_t)$ admits a uniform approximation error bound

$$\widehat{\Delta} = \sup_{x \in \mathcal{X},\, t \ge 1} \Big( \big|\widehat{\mu}_t(x) - \mu_t(x)\big| + \big|\widehat{\sigma}_t(x) - \sigma_t(x)\big| \Big) < \infty,$$

which monotonically decreases with $|\mathcal{S}|$ under standard Nyström bounds. (iv) Encrypted aggregation reproduces the plaintext global summary exactly (homomorphic exactness). (v) The information gain functions $\gamma_t^{\mathrm{g}}$ and $\gamma_t^{(i)}$ for the global and local models are concave functions of $t$. (This holds for common kernels like linear, SE, and Matérn.)

### C.2  Confidence Event for Exact GP-UCB

**Lemma 6** (GP-UCB Confidence Bound). *Under Assumption 5(i)-(ii), there exists a nondecreasing sequence $\beta_t$ such that, with probability at least $1 - \delta$, the event*

$$\mathcal{E} = \Big\{ \forall t \ge 1,\ \forall x \in \mathcal{X} :\ |f(x) - \mu_{t-1}(x)| \le \sqrt{\beta_t}\, \sigma_{t-1}(x) \Big\}$$

*holds.*

*Proof.* This is the standard GP-UCB confidence result over RKHS functions with sub-Gaussian noise. One constructs $\beta_t$ to simultaneously upper-bound the self-normalized concentration of the martingale differences $\{\epsilon_s\}$ and the RKHS prior deviation (see e.g., self-normalized process inequalities). The exact expression of $\beta_t$ is immaterial for our big-$\mathcal{O}$ bounds, it suffices that $\beta_t$ grows logarithmically with $t$ and $1/\delta$ so that $\mathbb{P}(\mathcal{E}) \ge 1 - \delta$. $\qquad\square$

### C.3  Exactness of Homomorphic Aggregation

**Lemma 7** (Homomorphic Exactness). *Let $\oplus$ be ciphertext-domain operations that form a ring homomorphism w.r.t. plaintext addition (for local summaries aggregation). If local summaries are encrypted and aggregated using $\oplus$ and then decrypted, the resulting plaintext equals the aggregation of plaintext local summaries (up to negligible modular wrap-around probability under standard parameter choices). Hence, the encrypted pipeline induces the same posterior as plaintext computation.*

*Proof.* By ring-homomorphism, for plaintexts $a, b$ with encryptions $\mathsf{Enc}(a), \mathsf{Enc}(b)$, we have $\mathsf{Dec}(\mathsf{Enc}(a) \oplus \mathsf{Enc}(b)) = a + b$. Federated GP summaries (means/covariances or their sufficient statistics) are polynomials in local contributions, hence, the decrypted aggregated ciphertext equals the corresponding polynomial evaluated on plaintexts. With parameters ensuring no overflow (large modulus and noise budget), decryption failure probability is negligible, thus the posterior matches the plaintext posterior exactly in $\mathbb{R}$ for the analysis. $\qquad\square$

## C.4 SPARSE-UCB REGRET

**Lemma 8** (Sparse-UCB Regret). *Run GP-UCB using the* sparse *posterior* $(\widehat{\mu}_{t-1}, \widehat{\sigma}_{t-1})$ *with the same exploration schedule* $\beta_t$ *as in Lemma 6. Under Assumption 5, on the event $\mathcal{E}$ of Lemma 6 the cumulative regret satisfies*

$$\widehat{R}_T \;=\; \sum_{t=1}^{T} \big(f(x^\star) - f(x_t)\big) \;=\; \mathcal{O}\Big(\sqrt{T\,\beta_T\,\gamma_T} \;+\; T\,\sqrt{\beta_T}\widehat{\Delta}\Big).$$

*Proof.* **Step 1 (Upper-Lower Bounds on $f(x)$).** On $\mathcal{E}$, for any $x$ and $t$,

$$|f(x) - \widehat{\mu}_{t-1}(x)| \;\leq\; |f(x) - \mu_{t-1}(x)| + |\mu_{t-1}(x) - \widehat{\mu}_{t-1}(x)| \;\leq\; \sqrt{\beta_t}\,\sigma_{t-1}(x) + \widehat{\Delta}.$$

Further, $\sigma_{t-1}(x) \leq \widehat{\sigma}_{t-1}(x) + |\sigma_{t-1}(x) - \widehat{\sigma}_{t-1}(x)| \leq \widehat{\sigma}_{t-1}(x) + \widehat{\Delta}$. Hence

$$|f(x) - \widehat{\mu}_{t-1}(x)| \;\leq\; \sqrt{\beta_t}\big(\widehat{\sigma}_{t-1}(x) + \widehat{\Delta}\big) + \widehat{\Delta} \;=\; \sqrt{\beta_t}\,\widehat{\sigma}_{t-1}(x) + \underbrace{\big(\sqrt{\beta_t} + 1\big)\widehat{\Delta}}_{\triangleq\, c_t\widehat{\Delta}}.$$

Thus we obtain both upper and lower bounds on $f(x)$ as follows.

$$f(x) \leq \widehat{\mu}_{t-1}(x) + \sqrt{\beta_t}\,\widehat{\sigma}_{t-1}(x) + c_t\widehat{\Delta},$$
$$f(x) \geq \widehat{\mu}_{t-1}(x) - \sqrt{\beta_t}\,\widehat{\sigma}_{t-1}(x) - c_t\widehat{\Delta}.$$

**Step 2 (Instantaneous Regret).** Let $x_t \in \arg\max_{x \in \mathcal{X}} \widehat{\mu}_{t-1}(x) + \sqrt{\beta_t}\,\widehat{\sigma}_{t-1}(x)$. Then

$$
\begin{aligned}
r_t &\triangleq\; f(x^\star) - f(x_t) \\
&\leq\; \Big[\widehat{\mu}_{t-1}(x^\star) + \sqrt{\beta_t}\,\widehat{\sigma}_{t-1}(x^\star) + c_t\widehat{\Delta}\Big] \;-\; \Big[\widehat{\mu}_{t-1}(x_t) - \sqrt{\beta_t}\,\widehat{\sigma}_{t-1}(x_t) - c_t\widehat{\Delta}\Big] \\
&=\; \underbrace{\Big(\widehat{\mu}_{t-1}(x^\star) + \sqrt{\beta_t}\,\widehat{\sigma}_{t-1}(x^\star) - \widehat{\mu}_{t-1}(x_t) - \sqrt{\beta_t}\,\widehat{\sigma}_{t-1}(x_t)\Big)}_{\leq\, 0 \text{ by choice of } x_t} \;+\; 2c_t\widehat{\Delta} \\
&\leq\; 2c_t\widehat{\Delta} \;=\; 2\big(\sqrt{\beta_t} + 1\big)\widehat{\Delta}.
\end{aligned}
$$

But we can also get a tighter bound by not fully canceling the UCB terms:

$$
\begin{aligned}
r_t &\leq \Big[\widehat{\mu}_{t-1}(x^\star) + \sqrt{\beta_t}\,\widehat{\sigma}_{t-1}(x^\star) + c_t\widehat{\Delta}\Big] - f(x_t) \\
&\leq \Big[\widehat{\mu}_{t-1}(x_t) + \sqrt{\beta_t}\,\widehat{\sigma}_{t-1}(x_t) + c_t\widehat{\Delta}\Big] - f(x_t) \quad \text{(by choice of } x_t) \\
&\leq \Big[\widehat{\mu}_{t-1}(x_t) + \sqrt{\beta_t}\,\widehat{\sigma}_{t-1}(x_t) + c_t\widehat{\Delta}\Big] - \Big[\widehat{\mu}_{t-1}(x_t) - \sqrt{\beta_t}\,\widehat{\sigma}_{t-1}(x_t) - c_t\widehat{\Delta}\Big] \\
&= 2\sqrt{\beta_t}\,\widehat{\sigma}_{t-1}(x_t) + 2c_t\widehat{\Delta}.
\end{aligned}
$$

Thus, $r_t \leq 2\sqrt{\beta_t}\,\widehat{\sigma}_{t-1}(x_t) + 2\big(\sqrt{\beta_t} + 1\big)\widehat{\Delta}$.

**Step 3 (Cumulative Regret).** Summing and using $\beta_t \leq \beta_T$ for $t \leq T$ (since $\beta_t$ is nondecreasing), we bound

$$\sum_{t=1}^{T} r_t \;\leq\; 2\sqrt{\beta_T}\sum_{t=1}^{T}\widehat{\sigma}_{t-1}(x_t) \;+\; 2\big(\sqrt{\beta_T} + 1\big)T\widehat{\Delta}.$$

We relate $\widehat{\sigma}$ to $\sigma$ via $\widehat{\sigma}_{t-1}(x_t) \leq \sigma_{t-1}(x_t) + \widehat{\Delta}$, giving $\sum_{t=1}^{T}\widehat{\sigma}_{t-1}(x_t) \leq \sum_{t=1}^{T}\sigma_{t-1}(x_t) + T\widehat{\Delta}$. Thus,

$$\sum_{t=1}^{T} r_t \leq 2\sqrt{\beta_T}\sum_{t=1}^{T}\sigma_{t-1}(x_t) + 2\sqrt{\beta_T}T\widehat{\Delta} + 2\big(\sqrt{\beta_T} + 1\big)T\widehat{\Delta} = 2\sqrt{\beta_T}\sum_{t=1}^{T}\sigma_{t-1}(x_t) + CT\sqrt{\beta_T}\widehat{\Delta}.$$

The standard GP information-gain argument implies $\sum_{t=1}^{T}\sigma_{t-1}(x_t) = \mathcal{O}(\sqrt{T\,\gamma_T})$ Srinivas et al. (2012). Therefore,

$$\widehat{R}_T \;=\; \sum_{t=1}^{T} r_t \;=\; \mathcal{O}\Big(\sqrt{\beta_T}\,\sqrt{T\,\gamma_T} \;+\; T\sqrt{\beta_T}\,\widehat{\Delta}\Big) \;=\; \mathcal{O}\Big(\sqrt{T\,\beta_T\,\gamma_T} \;+\; T\sqrt{\beta_T}\,\widehat{\Delta}\Big). \qquad \square$$

## C.5 REGRET DECOMPOSITION FOR LOCAL/GLOBAL MIXING

**Lemma 9** (Regret Decomposition). *Let $\mathcal{I}_T^{\mathrm{g}} = \{t \leq T : \text{the decision uses the } \text{global } \text{sparse posterior}\}$ and $\mathcal{I}_T^{\ell} = \{t \leq T : \text{the decision uses the } \text{local } \text{sparse posterior of some client } i\}$. Denote $T_{\mathrm{g}} = |\mathcal{I}_T^{\mathrm{g}}|$ and $T_{\ell}^{(i)}$ the number of local steps at client $i$. Then*

$$R_T \leq \widehat{R}_{T_{\mathrm{g}}}^{\mathrm{g}} + \sum_{i=1}^{M} \widehat{R}_{T_{\ell}^{(i)}}^{(i)},$$

*where each term admits a bound of the form in Lemma 8 with its corresponding information gain, i.e.,*

$$\widehat{R}_{T_{\mathrm{g}}}^{\mathrm{g}} = \mathcal{O}\Big(\sqrt{T_{\mathrm{g}}\,\beta_T\,\gamma_{T_{\mathrm{g}}}^{\mathrm{g}}} + T_{\mathrm{g}}\sqrt{\beta_T}\,\widehat{\Delta}\Big), \qquad \widehat{R}_{T_{\ell}^{(i)}}^{(i)} = \mathcal{O}\Big(\sqrt{T_{\ell}^{(i)}\,\beta_T\,\gamma_{T_{\ell}^{(i)}}^{(i)}} + T_{\ell}^{(i)}\sqrt{\beta_T}\,\widehat{\Delta}\Big).$$

*Taking expectations w.r.t. the mixing policy yields*

$$\mathbb{E}[R_T] = \mathcal{O}\Big(\sqrt{\mathbb{E}[T_{\mathrm{g}}]\,\beta_T\,\gamma_{\mathbb{E}[T_{\mathrm{g}}]}^{\mathrm{g}}} + \mathbb{E}[T_{\mathrm{g}}]\sqrt{\beta_T}\,\widehat{\Delta} + \sum_{i=1}^{M} \sqrt{\mathbb{E}[T_{\ell}^{(i)}]\,\beta_T\,\gamma_{\mathbb{E}[T_{\ell}^{(i)}]}^{(i)}} + \mathbb{E}[T_{\ell}^{(i)}]\sqrt{\beta_T}\,\widehat{\Delta}\Big).$$

*Proof.* **Step 1 (Local/Global Partition).** Decompose $R_T = \sum_{t \in \mathcal{I}_T^{\mathrm{g}}} r_t + \sum_{t \in \mathcal{I}_T^{\ell}} r_t$. On $\mathcal{I}_T^{\mathrm{g}}$ the algorithm uses the *global* sparse posterior and on $\mathcal{I}_T^{\ell}$, it uses a *local* sparse posterior at some client $i$.

**Step 2 (Apply Lemma 8 to each partition).** Treat the subsequence of global steps as a length-$T_{\mathrm{g}}$ sparse GP-UCB run whose information gain is $\gamma_{T_{\mathrm{g}}}^{\mathrm{g}}$ and analogously, for each $i$, treat the local steps as a length-$T_{\ell}^{(i)}$ run with information gain $\gamma_{T_{\ell}^{(i)}}^{(i)}$. Applying Lemma 8 to each subsequence yields the two given bounds for global and local respectively.

**Step 3 (Expectation over mixing).** If the mixing policy picks local vs. global at round $t$ with probabilities $p_t$ and $1 - p_t$ respectively, then $\mathbb{E}[T_{\mathrm{g}}] = \sum_{t=1}^{T}(1 - p_t)$ and $\mathbb{E}[T_{\ell}^{(i)}]$ equals the expected number of local steps at client $i$. By Assumption 5, the function $x \mapsto \sqrt{x\gamma_x}$ is concave. Therefore, using Jensen's inequality and the monotonicity of $\gamma$.,

$$\mathbb{E}\Big[\sqrt{T_{\mathrm{g}}\,\gamma_{T_{\mathrm{g}}}^{\mathrm{g}}}\Big] \leq \sqrt{\mathbb{E}[T_{\mathrm{g}}]\,\gamma_{\mathbb{E}[T_{\mathrm{g}}]}^{\mathrm{g}}}, \qquad \mathbb{E}\Big[\sqrt{T_{\ell}^{(i)}\,\gamma_{T_{\ell}^{(i)}}^{(i)}}\Big] \leq \sqrt{\mathbb{E}[T_{\ell}^{(i)}]\,\gamma_{\mathbb{E}[T_{\ell}^{(i)}]}^{(i)}}.$$

Linear terms pass expectation directly. This proves the claimed expectation bound. $\square$

## C.6 MAIN THEOREM

**Theorem 10** (FBO-FedGP Sublinear Cumulative Regret). *Under Assumption 5 and a mixing schedule $(p_t)_{t \geq 1} \subset (0, 1)$ with $\sum_{t=1}^{\infty}(1 - p_t) = \infty$ and $\sum_{t=1}^{\infty}(1 - p_t)^2 < \infty$, the FBO-FedGP with GP-UCB acquisition satisfies*

$$R_T = \mathcal{O}\Big(\sqrt{T\,\beta_T\,\Gamma_T} + T\sqrt{\beta_T}\,\widehat{\Delta}\Big), \qquad \Gamma_T \triangleq \gamma_{\mathbb{E}[T_{\mathrm{g}}]}^{\mathrm{g}} + \sum_{i=1}^{M} \gamma_{\mathbb{E}[T_{\ell}^{(i)}]}^{(i)}.$$

*In particular, $R_T/T \to 0$ and $r_T \to 0$ in probability as $T \to \infty$ provided $\widehat{\Delta} = o(1/\sqrt{\beta_T})$ or $T\sqrt{\beta_T}\,\widehat{\Delta} = o(T)$.*

*Proof.* **Step 1 (No error due to HE).** By Lemma 7, the encrypted aggregation reproduces the plaintext global summary, hence the only deviation from the exact centralized GP comes from GP approximation, captured by $\widehat{\Delta}$.

**Step 2 (Decompose regret).** Apply Lemma 9 and take expectations:

$$\mathbb{E}[R_T] = \mathcal{O}\Big(\sqrt{\mathbb{E}[T_{\mathrm{g}}]\,\beta_T\,\gamma_{\mathbb{E}[T_{\mathrm{g}}]}^{\mathrm{g}}} + \mathbb{E}[T_{\mathrm{g}}]\sqrt{\beta_T}\,\widehat{\Delta} + \sum_{i=1}^{M} \sqrt{\mathbb{E}[T_{\ell}^{(i)}]\,\beta_T\,\gamma_{\mathbb{E}[T_{\ell}^{(i)}]}^{(i)}} + \mathbb{E}[T_{\ell}^{(i)}]\sqrt{\beta_T}\,\widehat{\Delta}\Big).$$

**Step 3 (Role of Mixing Schedule).** The conditions on the mixing schedule $(p_t)$ ensure that $\mathbb{E}[T_{\mathrm{g}}] = \sum_{t=1}^{T}(1 - p_t) \to \infty$ as $T \to \infty$. This is necessary to ensure the global model's information gain $\gamma_{\mathbb{E}[T_{\mathrm{g}}]}^{\mathrm{g}}$ eventually grows, allowing the regret from global steps to be sublinear.

**Step 4 (Grouping Local/Global Terms).** Since each round is either global or local, $\sum_{m=1}^{M} \mathbb{E}[T_{\ell}^{(m)}] + \mathbb{E}[T_{\mathrm{g}}] = T$. Define $\Gamma_T$ as in the theorem. Using $\sqrt{a} + \sqrt{b} \leq \sqrt{2(a+b)}$, we group the square-root terms to obtain

$$\mathbb{E}[R_T] = \mathcal{O}\Big(\sqrt{T\,\beta_T\,\Gamma_T} + T\sqrt{\beta_T}\,\widehat{\Delta}\Big).$$

**Step 5 (Sublinearity).** For common kernels (such as SE), $\gamma_t = o(t)$, hence $\Gamma_T = o(T)$. Taking $\beta_T = O(\log T)$ and using $\Gamma_T = o(T)$ yields $\sqrt{T\,\beta_T\,\Gamma_T} = o(T)$. If $\widehat{\Delta} = o(1/\sqrt{\beta_T})$ or at least $T\sqrt{\beta_T}\,\widehat{\Delta} = o(T)$, then $R_T/T \to 0$. Standard arguments (e.g., from sublinear cumulative regret on bounded domains) imply $r_T \to 0$ in probability. $\qquad\square$

### C.7 COROLLARIES

**Exact GP as a special case.** If $\widehat{\Delta} = 0$ (exact GP, or sparse approximation sufficiently equal to exact), the bound reduces to $R_T = \mathcal{O}\big(\sqrt{T\,\beta_T\,\Gamma_T}\big)$, matching the exact GP-UCB rate under mixing of global/local GPs.

**Controlling $\widehat{\Delta}$.** Under standard Nystrom/FITC/PITC error bounds, $\widehat{\Delta} = \mathcal{O}(|\mathcal{S}|^{-1/2})$ for a support set $\mathcal{S}$ of size $|\mathcal{S}|$. To ensure the approximation error term is sublinear, we require $T\sqrt{\beta_T}\widehat{\Delta} = T\sqrt{\beta_T} \cdot \mathcal{O}(|\mathcal{S}|^{-1/2}) = o(T)$. This holds if $\sqrt{\beta_T}\widehat{\Delta} = o(1)$, or equivalently, $\widehat{\Delta} = o(1/\sqrt{\beta_T})$. Since $\beta_T = \mathcal{O}(\log T)$, we need $\widehat{\Delta} = o(1/\sqrt{\log T})$. A practical strategy is to let the inducing set size grow with $T$, for example, choosing $|\mathcal{S}| = \Theta((\log T)^{1+\epsilon})$ for some $\epsilon > 0$, which ensures $|\mathcal{S}|^{-1/2} = o(1/\sqrt{\log T})$. Alternatively, choosing $|\mathcal{S}| = \Theta(T^\rho)$ for any $\rho > 0$ is also sufficient, as then $T\sqrt{\log T} \cdot T^{-\rho/2} = T^{1-\rho/2}\sqrt{\log T} = o(T)$.

**Practical guidance for choosing $|\mathcal{S}|$.** In practice one must trade off the approximation benefits of a larger inducing set against the communication and computation costs (each client uploads $\mathcal{O}(|\mathcal{S}|^2)$ summary elements and the server performs $\mathcal{O}(|\mathcal{S}|^3)$ central linear-algebra work). For small BO budgets (e.g., $T \lesssim 10^2$) we find $|\mathcal{S}| \approx$ 6–12 (e.g., 8) offers a good balance, while for medium budgets ($10^2 \lesssim T \lesssim 10^4$) a choice around $|\mathcal{S}| \approx$ 12–32 works well. For instance $|\mathcal{S}| \approx \lceil (\log T)^{1.25} \rceil$ yields $|\mathcal{S}| \approx 11$ at $T = 10^3$ and $|\mathcal{S}| \approx 16$ at $T = 10^4$ and for very large budgets ($T \gtrsim 10^4$) using $|\mathcal{S}| \approx$ 32–64 or a slowly growing polynomial rule $|\mathcal{S}| = \lceil T^\rho \rceil$ with a small $\rho \in (0, 0.25)$ is sufficient to make $\widehat{\Delta} = o(1/\sqrt{\log T})$ while keeping costs manageable. If the intrinsic dimension $d$ is large, scale $|\mathcal{S}|$ proportionally to the intrinsic dimension (for example multiply the above heuristics by $\sqrt{d/d_0}$ where $d_0$ is a chosen baseline). A simple and effective adaptive policy is to start with a small $|\mathcal{S}| = S_{\min}$ (e.g., 8 or 12) and periodically increase it by a small increment (e.g., $+4$ or $+8$) every $K$ iterations (e.g., $K \approx \sqrt{T}$ or another budget-proportional schedule). This amortizes the cubic cost of resizing while ensuring the approximation error decays over the run. For concrete cost mappings, Table 2 reports wall-clock times for $|\mathcal{S}| \in \{8, 16, 32\}$ and $M \in \{10, 20, 30\}$, which can be used to calibrate the above rules-of-thumb for a target runtime or communication budget.

## D SUPPLEMENTARY EXPERIMENTAL RESULTS

### D.1 GLOBAL-ONLY VS. LOCAL-ONLY SURROGATES

We analyze the contribution of both global and local components by evaluating ablated versions of FBO-FedGP. Figure 3(d–f) shows that the global-only variant achieves reasonable performance (60-80% accuracy) but converges slower than the full framework, particularly on heterogeneous tasks like in-hospital mortality prediction ( 40% accuracy). This indicates that while the global model captures the overall landscape, it lacks client-specific adaptation.

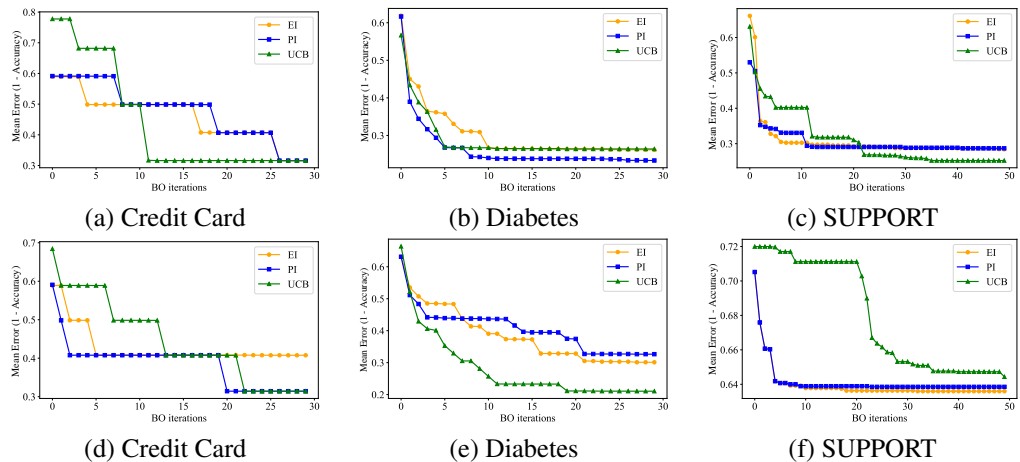

Figure 3: Convergence of FBO-FedGP on credit card default, diabetes and in-hospital mortality classification tasks using local-only (top row) and global-only (bottom-row) surrogate variant.

Table 2: Computation times on the DIABETES dataset when varying the number of clients and support set size. Reported times are in seconds.

| No. of Clients | $|\mathcal{S}| = 8$ | $|\mathcal{S}| = 16$ | $|\mathcal{S}| = 32$ |
|---|---|---|---|
| 10 | 167.36 | 174.56 | 198.03 |
| 20 | 391.31 | 397.30 | 473.30 |
| 30 | 607.15 | 619.53 | 659.17 |

Conversely, the local-only variant (Figure 3(a–c)) performs well on highly heterogeneous data (matching full FBO-FedGP on mortality prediction) but underperforms on moderately heterogeneous tasks ($70\%$ vs. $> 80\%$ for credit default). This demonstrates that local models excel at client adaptation but lack global context.

These results confirm that both components provide complementary benefits: the global surrogate enables broad exploration, while local surrogates allow for client-specific refinement. Their combination in FBO-FedGP yields superior performance across diverse federated settings.

### D.2 EFFECT OF FEDERATION SIZE AND ACQUISITION STRATEGY

Figure 4 shows performance on the in-hospital mortality task with varying client counts (3, 6, 9). With 3 clients, GP-UCB significantly outperforms EI and PI, suggesting exploration-focused acquisition is beneficial in small federations. With 6 clients, performance differences diminish, and with 9 clients, all acquisition functions converge to nearly identical performance. This indicates that larger federations naturally stabilize the optimization process through increased data diversity, reducing sensitivity to acquisition function choice.

### D.3 RANDOM SUPPORT SET SELECTION

Figure 6 reports the optimization behavior of FBO–FedGP when the support set $\mathcal{S}$ is chosen uniformly at random from the input domain with three different sizes, $|\mathcal{S}| \in \{8, 16, 32\}$, across four real-world classification tasks. The plots show mean error (1 - accuracy) versus BO iterations for (a) Credit-card, (b) Diabetes, (c) SUPPORT (in-hospital mortality), and (d) Landmines. The results show that the optimization is considerably slower on credit-card, SUPPORT and landmines classification task compared to the performance reported in Fig. 2. These results show that informed support set selection is an important parameter to enhance the fidelity of the surrogate dGP model to achieve better optimization performance.

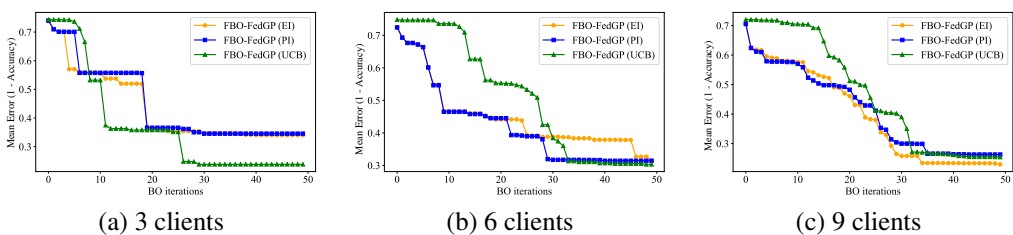

(a) 3 clients        (b) 6 clients        (c) 9 clients

Figure 4: Performance of FBO-FedGP with 3, 6 and 9 clients on SUPPORT dataset.

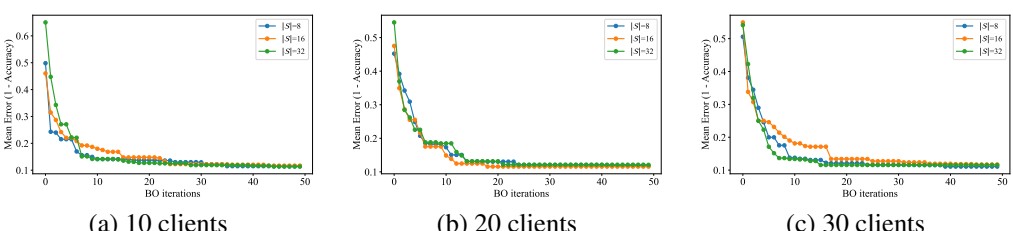

(a) 10 clients        (b) 20 clients        (c) 30 clients

Figure 5: Performance of FBO-FedGP with 10, 20 and 30 clients on Diabetes dataset.

With randomly chosen support-set across the four benchmarks, the following observations emerge. First, increasing the support-set size generally improves convergence speed and final accuracy, but the magnitude of this improvement depends strongly on the dataset. The SUPPORT dataset (highly heterogeneous clinical data) exhibits the largest sensitivity to $|\mathcal{S}|$, i.e, the smallest support ($|\mathcal{S}| = 8$) stagnates at substantially higher error, while $|\mathcal{S}| = 16$ and $|\mathcal{S}| = 32$ reduce error rapidly and reach much lower final values. This behavior indicates that heterogeneous data benefit from larger and more representative support sets because more support points are needed to capture distinct local modes and to produce a reliable sparse GP posterior. By contrast, the Diabetes and Landmines tasks show only minor differences between the three support sizes. All schedules converge to nearly the same low error within the early iterations, suggesting that these datasets either contain sufficiently dense information that a small support already provides an adequate surrogate, or that the problem structure is easier to approximate with sparse representations. For these tasks the gains from increasing $|\mathcal{S}|$ are marginal, implying that modest support budgets (e.g., $|\mathcal{S}| = 16$) strike a favorable balance between surrogate quality and resource cost.

### D.4 Effect of Number of Clients and Support Set Size

To evaluate the scalability of FBO-FedGP, we vary both the number of participating clients ($M \in \{10, 20, 30\}$) and the support set size ($|\mathcal{S}| \in \{8, 16, 32\}$) on the DIABETES dataset. Figure 5 reports the convergence behavior in terms of classification loss over BO iterations. The results show that the method maintains stable convergence as the number of clients increases, with only a modest slowdown for larger $M$. Larger support sets (e.g., $|\mathcal{S}| = 32$) generally lead to faster convergence and slightly improved predictive accuracy, reflecting the trade-off between communication cost and model fidelity. These findings empirically support the theoretical analysis (Sec. 4) that the error term $\hat{\Delta}$ decreases as the support set size grows.

To complement these convergence results, Table 2 reports the corresponding wall-clock computation times. As expected, the overall runtime increases with both the number of clients and the support set size. For instance, moving from $M = 10$ to $M = 30$ roughly triples the computation time, while increasing the support set from $|\mathcal{S}| = 8$ to $|\mathcal{S}| = 32$ incurs only a moderate additional cost. Together with Figure 5, these results highlight the practical trade-off, i.e., larger support sets improve convergence speed and accuracy at the expense of slightly higher runtime, while the method scales well with the number of clients.

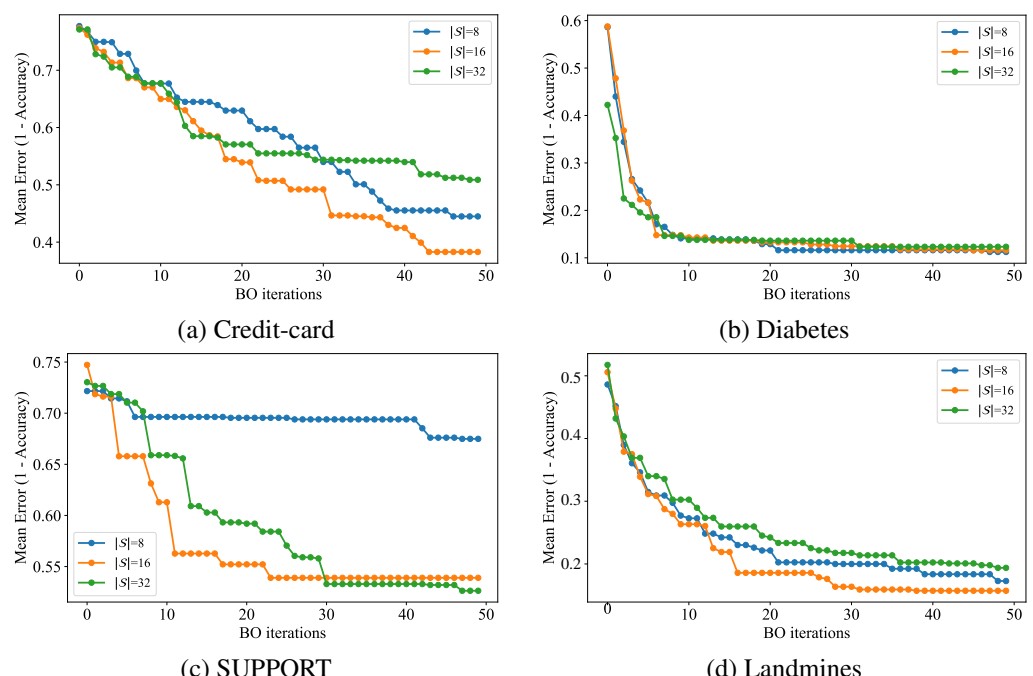

(a) Credit-card

(b) Diabetes

(c) SUPPORT

(d) Landmines

Figure 6: Performance of FBO-FedGP with randomly selected support set points from input domain.

Table 3: Computation times (in seconds) for credit-card, diabetes, SUPPORT and landmines datasets with 30, 30, 9 and 29 client setup respectively over 50 BO iterations.

| Dataset | $|\mathcal{S}| = 8$ | $|\mathcal{S}| = 16$ | $|\mathcal{S}| = 32$ |
|---|---|---|---|
| Credit-card | 453.59 | 544.69 | 564.22 |
| Diabetes | 453.29 | 481.36 | 636.44 |
| Support | 153.74 | 157.74 | 169.53 |
| Landmines | 416.83 | 433.25 | 459.90 |

## D.5 ANALYSIS OF MIXING SCHEDULES

Figure 7 and Table 4 compare four mixing schedules ($p_t = 1 - 1/t, 1 - 1/\sqrt{t}, 1 - 1/t^{0.25}, 1 - 1/t^{0.75}$) across three tasks.

All schedules achieve similar final performance, but with different computational costs and convergence speeds. The slower-decaying schedule ($1 - 1/t^{0.25}$) emphasizes global updates, achieving rapid initial improvement but at the highest computational cost (e.g., 246.33s vs 172.06s for credit task). The faster-decaying schedule ($1 - 1/t^{0.75}$) reduces computation time but may plateau earlier.

For practical deployment, $1 - 1/\sqrt{t}$ provides the best balance, offering competitive convergence with moderate computational requirements (137.75s for mortality task). These results demonstrate that while schedule choice affects efficiency, FBO-FedGP is robust to different mixing strategies.

## D.6 VISUALIZATION OF EXPLORED POINTS IN OPTIMIZATION

To further examine the behavior of the proposed FBO-FedGP framework, the explored points are visualized across various synthetic benchmark functions, as shown in Figure 8. Each subplot corresponds to one of the benchmark functions considered in our evaluation. On the Shifted Sphere function (F1), the trajectory of explored points demonstrates smooth convergence toward the shifted global optimum, confirming the capability of FBO-FedGP to adapt even in simple unimodal landscapes. For the Shifted Double Sum function (F2), the optimizer progressively moves toward the

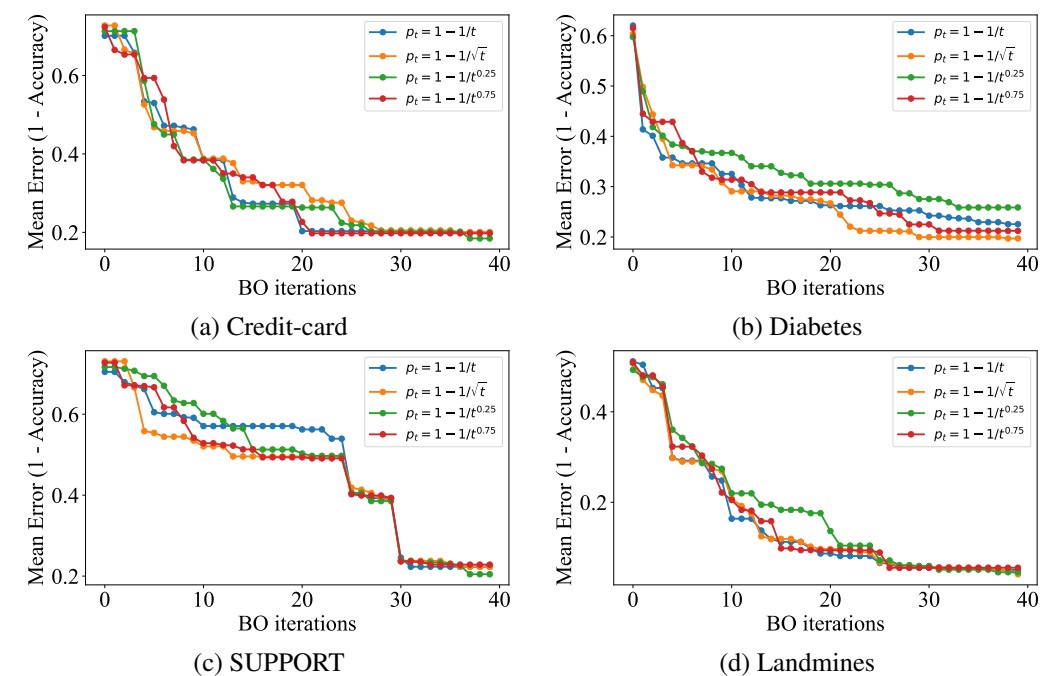

Figure 7: Convergence of FBO-FedGP with varying $p_t$ schedules for local/global mixing of surrogate models.

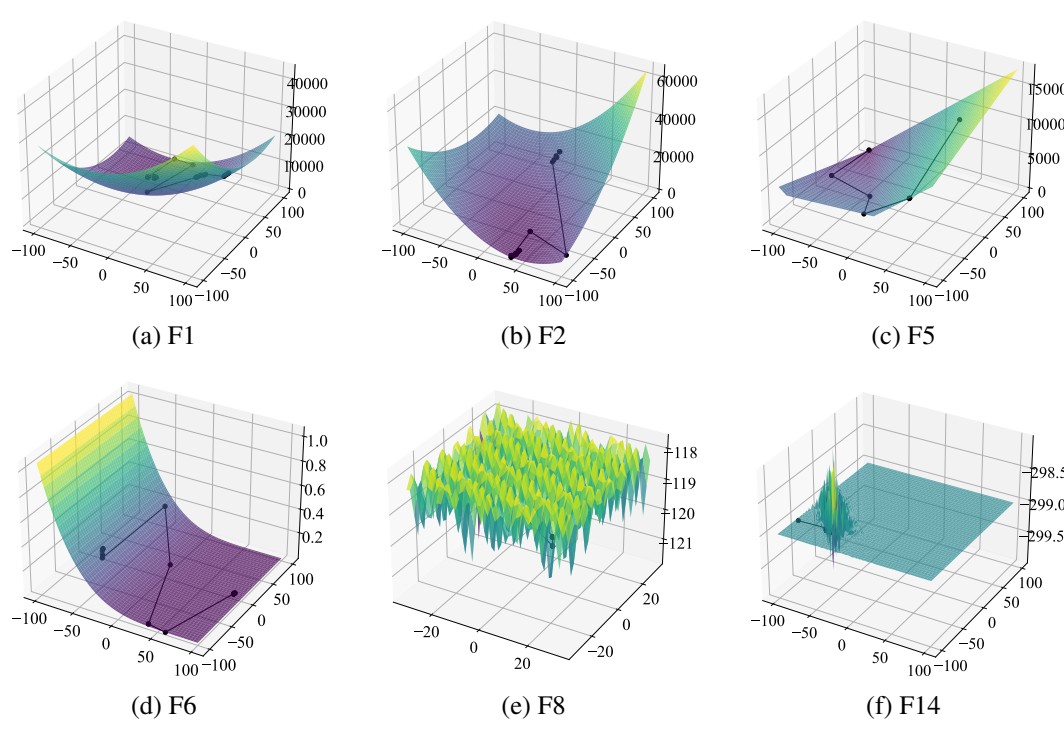

Figure 8: Visualization of explored points by FBO-FedGP in optimization.

optimum while adjusting to the cumulative dependency of variables, reflecting its ability to handle structured search spaces.

Table 4: Computation times on credit-card, diabetes and in-hospital mortality classification using different $p_t$ policy schedules for local/global model mixing. (time in seconds.)

| Dataset | $p_t = 1 - 1/t$ | $p_t = 1 - 1/\sqrt{t}$ | $p_t = 1 - 1/t^{0.25}$ | $p_t = 1 - 1/t^{0.75}$ |
|---|---|---|---|---|
| Credit-card | 156.11 | 185.39 | 246.33 | 172.06 |
| Diabetes | 421.06 | 527.67 | 705.90 | 543.12 |
| In-hospital Mortality | 110.57 | 137.75 | 193.87 | 120.45 |
| Landmines | 322.85 | 435.28 | 605.64 | 350.74 |

Table 5: Paired Wilcoxon signed-rank tests comparing FBO-FedGP to baselines (two-sided). $N = 20$ benchmark problems, per-problem values are the mean over 10 seeds. P-values are Holm–Bonferroni adjusted. Effect size is the rank-biserial correlation (rb) computed as $(n_+ - n_-)/n$, where $n_+$ ($n_-$) are counts of problems where FBO-FedGP is worse (better) than the baseline. Negative effect sizes indicate FBO-FedGP has lower loss (better performance).

| Baseline | $W$ | Raw p-value | Adj. p-value | Effect size (rb) | Sig. |
|---|---|---|---|---|---|
| EPPBO | 9.00 | $6.294 \times 10^{-5}$ | $1.259 \times 10^{-4}$ | $-0.900$ | ** |
| FMTBO | 0.00 | $1.907 \times 10^{-6}$ | $9.537 \times 10^{-6}$ | $-1.000$ | ** |
| FTS | 0.00 | $1.907 \times 10^{-6}$ | $9.537 \times 10^{-6}$ | $-1.000$ | ** |
| FEPPBO | 0.00 | $1.907 \times 10^{-6}$ | $9.537 \times 10^{-6}$ | $-1.000$ | ** |
| FCSBO | 18.00 | $4.826 \times 10^{-4}$ | $4.826 \times 10^{-4}$ | $-0.800$ | ** |

'**' indicates significance at $\alpha = 0.05$ after Holm adjustment.

In the case of Schwefel's Problem 2.6 with the global optimum located on the bounds (F5), the explored points show that the optimizer is able to traverse the deceptive regions of the function and concentrate around the boundary optimum. For the Shifted Rosenbrock function (F6), the optimizer successfully navigates the narrow curved valley toward the shifted optimum, demonstrating robustness against ill-conditioning and shifted landscapes. The Shifted Rotated Ackley function with global optimum on the bounds (F8) presents a highly multimodal surface with many local minima, nevertheless, the explored points initially spread across different regions before converging toward the true optimum at the boundary, highlighting the exploration strength of the framework. Finally, for the Shifted Rotated Expanded Schaffer's function (F14), the optimization trajectory covers regions with repetitive rugged structures and high multimodality, yet avoids premature convergence, demonstrating resilience against oscillatory landscapes.

Overall, these visualizations highlight the adaptability of FBO-FedGP across functions of varying complexity. On simpler unimodal problems such as F1 and F2, convergence is rapid and direct, while on challenging multimodal and shifted functions such as F5, F6, F8, and F14, the optimizer demonstrates robustness by systematically exploring the landscape without being trapped in local minima. This balance between exploration and exploitation validates the suitability of FBO-FedGP for both smooth and rugged objective functions.

### D.7 STATISTICAL SIGNIFICANCE TESTING

We evaluate statistical significance using paired Wilcoxon signed-rank tests (two-sided) comparing FBO-FedGP (reference) against each baseline. All reported $p$-values are adjusted for multiple comparisons using the Holm–Bonferroni procedure. Tests were performed over $N = 20$ benchmark problems (paired comparisons). For each problem we aggregate results over 10 random seeds by reporting the mean performance per problem. The Wilcoxon test is then applied to these paired per-problem summaries. Results are summarized in Table 5.

All pairwise comparisons are significant after Holm adjustment. The (negative) rank-biserial effect sizes indicate that FBO-FedGP consistently attains lower classification loss than the baselines on the evaluated problems, with medium-to-large practical effect magnitudes.

