# OpenReview forum: "Federated Bayesian Optimization based on Secure Distributed Gaussian Processes"
_ICLR.cc/2026/Conference — ICLR 2026 Conference Withdrawn Submission_

### Official Review · Reviewer_4noF · 2025-10-27

**Soundness:** 2
**Presentation:** 1
**Contribution:** 2
**Rating:** 2
**Confidence:** 4

**Summary:**

The paper introduces FBO-FedGP, a privacy-preserving federated Bayesian optimization framework that combines multiparty homomorphic encryption with a distributed sparse Gaussian process surrogate. In FBO-FedGP, clients compute encrypted local summaries on a shared inducing set, the server homomorphically aggregates them into a global summary, and all parties collaboratively decrypt to obtain a global posterior. Furthermore, a one-shot federated clustering procedure yields compact, informative inducing points, and a monotonically increasing mixing schedule balances early global exploration with later local exploitation to address heterogeneity and communication costs. Finally, the authors prove sublinear cumulative regret for GP-UCB with bounds that explicitly tie performance to support-set approximation error, and demonstrate, across 20 CEC-2005 benchmarks and several real federated tasks, statistically significant improvements over five privacy-preserving BO baselines.

**Strengths:**

1. This work introduces a novel combination of multiparty homomorphic encryption (HE) with a distributed sparse Gaussian process (dGP) surrogate specifically for federated Bayesian optimization (FBO), avoiding the accuracy–privacy trade-offs inherent in DP-based or transform-based approaches.

2. The empirical results are good. The proposed FBO-FedGP outperforms other tested baselines in most cases.

**Weaknesses:**

1. The novelty of the proposed method is limited. Most components used in this paper, including the dGP, HE-based aggregation, and the mixture of local and global sampling strategy, are existing methods. The technical difficulty or challenge of combining these methods is not clearly discussed.

2. The additional communication and computational overhead introduced by homomorphic encryption is not analyzed either theoretically or empirically. The 128-bit security parameter used in the paper does not meet practical application requirements.

3. Appendix B only shows the security analysis of dGP. The security of the proposed FBO method is not clearly analyzed. For example, although the original data is not shared, the locally proposed support set and the decrypted global summary may inadvertently encode distributional characteristics of the local data.

4. The writing of this paper is poor. The main paper is not self-contained and includes many undefined notations. It is hard to follow without reading the appendix. For example,

* In line 127 and line 135, $\bar{b}$ and $\Sigma_{SS}$ are not defined;

* In Section 2, many terms such as "support set" and "local summary" are not well introduced, which makes it hard for a non-expert of sparse GP to follow the paper.

* $T_g$ and $T_l$ in Theorem 3 are not defined.

* In Assumption 1, it's not clear how the infinite bound of $\hat{\Delta}$ is related to the "monotonically decrease" property of the sparse posterior error.

**Questions:**

1. Can you provide a detailed breakdown of wall-clock time across stages (local GP summary computation, encryption, server aggregation, decryption, acquisition optimization) and how each scales with clients M, input dimension d, and support set size |S|?

2. The global summary calculation in (2) involves matrix inversion. How can this be accomplished in a homomorphic encryption environment? If approximate calculations are used, additional error analysis is required compared to the exact plaintext calculations.

3. In Theorem 3, the regret $R_T$ is proved to be sub-linear in $T$ only if the approximation error $\hat{\Delta}$ is bounded by an equation of $\beta$, which is inconsistent with the bound in Assumption 1. How can you guarantee that the bound of $\hat{\Delta}$ holds? How is this bound related to $|S|$ and the support set selection algorithm?

---

### Official Review · Reviewer_8J2q · 2025-10-29

**Soundness:** 2
**Presentation:** 2
**Contribution:** 2
**Rating:** 2
**Confidence:** 2

**Summary:**

The paper focuses on distributed Bayesian optimization in federated learning (FL), proposing a method based on distributed sparse Gaussian processes combined with existing cryptographic methods to avoid having to centralize client data for Bayesian optimization. The proposed method comprises of two phases: i) an initial local and global clustering steps to find a set of inducing points shared globally between the clients and the server, and ii) the main FL optimization loop, which chooses between local and global updates based on a client-specific mixing schedule. The paper includes a theoretical convergence proof for the algorithm under some assumptions, as well as simulated and some real data experiments with various existing methods as baselines.

**Strengths:**

* The paper is mostly nicely written and easy to read (although I still have plenty of things to ask after reading this, see Questions).
* Implementation code is provided with the submission.
* There is a theoretical convergence analysis under some assumptions.
* The proposed method is compared against several existing baselines (although see Questions for some comments on this).

**Weaknesses:**

1) Many experimental and other details are missing (see Questions for more details).

2) The threat models and goals of some baselines are quite different from the proposed method (see Questions for more details).

**Questions:**

## In decreasing order of importance

1) Please add more experimental details, eg, which model(s) you use for real-world experiments, how many clients do you have for the real-world datasets, at least a short description in the Appendix about the baselines (including what parameter values such as privacy budget for DP you use, and how these were set for each), how is the reported accuracy calculated (is this mean over some local possibly heterogeneous testing sets, some centralized test set or something else), how many parameters are you optimizing for in each task, runtimes/complexity also for baselines.

2) I am not sure I understand how the mixing works in practice: if the choice of local steps vs global update is done separately by each client based on their personal mixing schedule, how does a given client know which other clients are included in the protocol, do you have some shared randomness, just do extra communications or something else? Do the other clients who chose to do local updates just wait while the global model is being updated by some other set of clients, or this is meant to be run only sequentially over clients?

3) Please include some variability measure like std or standard error of the mean for all the results.

4) A general comment on privacy vs security (related, eg, to the discussion about related work in the intro, see eg lines 66-67): please do not use privacy as a synonym for secure computation. The privacy and threat model in differential privacy (DP, avoid leaking too much information about the sensitive data) is different (and orthogonal) to what homomorphic encryption (HE) including this paper aims for (protect the computation without caring if the end result leaks everything in the input data). There is very little point, eg, in arguing that you should use HE instead of DP because DP costs utility while HE does not: you use something like DP if you want privacy guarantees for sensitive data, you use HE when you want to guarantee that computations do not leak information beyond the final result, these are not competing alternatives to reach the same goal (and I have no idea what kind of privacy you might mean on lines 60-62).

5) Related to the previous comment, it would be very good to be explicit about the differences in the threat models and aimed for protections between various methods in the experimental section, instead of presenting these as directly competing alternatives like in Table 1.

6) Fig 1, related to the previous 2 comments: I would not present the results with a single mean line over all baselines, as this can be pretty misleading (see, eg, F1 & F2); consider eg, plotting against best or best & 2nd best performing baselines, or against best within a given threat model.

7) Would be good to be a bit more explicit about the intended FL setting, eg, if this is cross-silo and not cross-device, how many clients and how much compute do you take etc.

8) Fig 6: can you please include the non-random support set result from Fig 2 here as well for ease of comparison?

9) Sec 6.2: it looks a bit weird that, eg, when the proposed method outperforms the best baseline by roughly 130 points difference in actual values for F8 (-120 vs 10) this is written as a substantial difference, but when a baseline beats the proposed method, eg, in F12 by more than 300 (460 vs 90), it is described as a competitive performance.

10) Lines 654-55: this seems like a really weird definition for robustness against missing or non-participating clients? I would expect that robustness here would be more like being able to decrypt and get in some sense a correct result even if some clients fail during the protocol run.

---

### Official Review · Reviewer_bo8F · 2025-10-30

**Soundness:** 2
**Presentation:** 3
**Contribution:** 1
**Rating:** 2
**Confidence:** 3

**Summary:**

This paper propose FBO-FedGP, a federated Bayesian optimization method that uses multi-party homomorphic encrypotion to aggregate clients' GP summary statistics over a shared inducing set, coupled with a distributed sparse GP surrogate and an increasing mixing schedule that alternates global and local acquisitions. It provides analysis for sublinear cumulative regret tied to the support-set approximation error, and reports experiments on 20 CEC benchmarks and several real tasks, along with an HE instantiation and measured aggregation error.

**Strengths:**

The proposed FBO-FedGP is a novel framewrok that is highly original, which moves beyond noisy DP-based approaches to offer strong cryptographic privacy. The design features a computationally feasible sparse GP surrogate and a theoretically-motivated mixing schedule to handle data heterogeneity. The paper provides a practical blueprint for secure, high-performance black-box optimization in sensitive and distributed applications. The method is nicely presented, making the complex interplay of BO, FL, and cryptography accessible.

**Weaknesses:**

# W1:

The introduction successfully explains why existing methods have drawbacks. They describe pain points with DP and with compressed sensing (exact GP is cubic, communication heavy). However, this paper does not turn this into a clear, specific, measurable target that the paper then explicitly aims for and measures itself against (e.g., at fixed privacy/bit-security and communication budget, achieve X). So, the gap stated is contextual, not nailed down as a testable goal.



# W2:

The theory assumes homomorphic exactness (Assumption 1-(iv)). However, the implementation uses CKKS, which is approximate. They also report small but non-zero errors. The analysis never carries these CKKS errors through to the posterior or regret. This creates a fundamental theory-implementation mismatch.



# W3:

The regret bound $R_{T}$ (Theorem 3) depends on a uniform sparse-approximation error $\hat{\Delta}$, but $\hat{\Delta}$ is only assumed finite and decaying with $|\mathcal{S}|$. There is no explicit rate in $T$, dimension $d$, kernel class, or under the inducing point selection (Alg. 1). There is also no link from $|\mathcal{S}|$ to the compute or communication budget.


While the schedule conditions guarantee infinitely many global syncs with bounded variance, the paper stops short of translating a concrete choice (e.g., $p_{t} = 1-\sqrt{t}$) into an explicit regret-vs-HS/communication cost trade-off. This is essential to justify the claimed "balance" and to tune the method under real budgets.


# W4:

The empirical verification is misplaced. That is, the claim "128-bit security" is presented as a conclusion from an empirical test, but that test is irrelevant tot he claim being made.

The paper states "When one share is intentionally withheld the attempted plaintext reconstruction exhibits modular
wrap-around and produces values with extremely large magnitudes... These values are astronomically large compared to original plaintext values, validating that the protocol outputs computationally meaningless data in the absence of all shares"

This demonstrates the **correctness and robustness** of the collaborative decryption protocol. It shows the system fails gracefully and securely if a client drops out. It is a functional text, **not a security evaluation**.

This experiment says nothing about the bit-security of the underlying RLWE problem. An adversary's goal is not to decrypt with a missing share. Instead, it is to break the encryption while the protocol is being followed correctly.

In addition, the paper lacks of a security proof or reference to standards. It presents parameters and simply asserts they provide 128-bit security without showing the work or referencing the tools that the cryptographic community uses to make such a determination.

The actualy security level depends entirely on the mathematical relationship between $N$, $q$, and $\sigma$ within the context of known lattice attacks, which has not been demonstrated.

**Questions:**

Please see the comments in Weaknesses.

---

### Official Review · Reviewer_K9pT · 2025-10-31

**Soundness:** 3
**Presentation:** 4
**Contribution:** 3
**Rating:** 6
**Confidence:** 2

**Summary:**

This paper proposes FBO-FedGP, a federated Bayesian optimization framework that combines homomorphic encryption with distributed sparse Gaussian processes. The approach uses federated clustering to select support sets, encrypts local GP summaries for secure aggregation, and employs an adaptive local/global mixing schedule. The authors provide theoretical regret analysis and demonstrate improvements over five baselines on 20 synthetic functions and four real-world tasks.

**Strengths:**

S1. Complete implementation of HE-based secure federated BO, addressing an important problem at the intersection of privacy and optimization.
S2. Comprehensive empirical evaluation with 24 tasks total, statistical significance testing (Wilcoxon + Holm correction), and thorough ablations on support set size, federation size, and mixing schedules.
S3. Strong experimental results showing consistent improvements over privacy-preserving baselines (EPPBO, FMTBO, FTS, FEPPBO, FCSBO) with statistically significant gains.
S4. End-to-end solution with concrete HE parameters (128-bit security), negligible numerical error (10^-7).

**Weaknesses:**

W1: The core technical component—secure distributed GP with homomorphic encryption—is directly from Nawaz et al. 2024. The main addition is using federated k-means for support set selection, but this requires clients to share cluster centers in plaintext (Algorithm 1, lines 3-4), creating a privacy vulnerability that has not been analyzed. The paper extensively analyzes encryption security (Appendix B) but completely ignores the information leakage from sharing cluster centers about data distribution.
W2: Algorithm 1 applies standard federated k-means (not novel) without theoretical analysis of: (1) why this greedy approach is appropriate for support set selection, (2) how clustering quality affects the approximation error, or (3) principled guidance for hyperparameter selection.
W3: Theorem 3's proof (Appendix C) mechanically combines: (1) GP-UCB bounds (Srinivas et al. 2012), (2) sparse GP approximation error (Mutny & Krause 2020), and (3) Robbins-Monro mixing decomposition. The claim "first regret bound for HE-based federated BO" is technically true but misleading since HE is homomorphic—the mathematics are identical to non-encrypted sparse GP bounds with trivial mixing decomposition.
W4: Collaborative decryption requires all M clients to provide secret shares (Eq. 14). If even one client drops out, the global update fails completely. This is highly impractical for federated settings with intermittent participation, stragglers, or voluntary engagement.

**Questions:**

Q1: Can you provide formal privacy analysis for sharing cluster centers? What is the information leakage, and have you considered differential privacy noise or secure multi-party clustering?
Q2: How does your theory differ from Mutny & Krause 2020 (sparse GP-UCB) beyond the mixing decomposition?
Q3: Can you compare federated k-means to other support selection methods (DPP, herding, gradient optimization) and provide theoretical or empirical justification for this choice?

---

### Note · Authors · 2026-04-14

I have read and agree with the venue's withdrawal policy on behalf of myself and my co-authors.

---

### Meta-Review · Area_Chair_faDd · 2025-12-08

**Summary:**

Majority of reviewers, including all the confident ones, recommend rejection. There is no author rebuttal.

**Reviewer Concerns:**

There is no rebuttal so all concerns remain outstanding.

**Reviewer Scores:**

No change.

---

### Decision · Program_Chairs · 2026-01-26

Reject